# Proteome-wide and matrisome-specific alterations during human pancreas development and maturation

Zihui Li [1,4], Daniel M. Tremmel [2,4], Fengfei Ma [3], Qinying Yu[3], Min Ma[3], Daniel G. Delafield [1], Yatao Shi[3], Bin Wang[3], Samantha A. Mitchell[2], Austin K. Feeney[2], Vansh S. Jain [2], Sara Dutton Sackett [2], Jon S. Odorico [2,5✉] & Lingjun Li [1,3,5✉]

The extracellular matrix (ECM) is unique to each tissue and capable of guiding cell differentiation, migration, morphology, and function. The ECM proteome of different developmental stages has not been systematically studied in the human pancreas. In this study, we apply mass spectrometry-based quantitative proteomics strategies using N,N-dimethyl leucine isobaric tags to delineate proteome-wide and ECM-specific alterations in four age groups: fetal (18-20 weeks gestation), juvenile (5-16 years old), young adults (21-29 years old) and older adults (50-61 years old). We identify 3,523 proteins including 185 ECM proteins and quantify 117 of them. We detect previously unknown proteome and matrisome features during pancreas development and maturation. We also visualize specific ECM proteins of interest using immunofluorescent staining and investigate changes in ECM localization within islet or acinar compartments. This comprehensive proteomics analysis contributes to an improved understanding of the critical roles that ECM plays throughout human pancreas development and maturation.

[1] Department of Chemistry, University of Wisconsin-Madison, Madison, WI, USA. [2] Department of Surgery, Division of Transplantation, School of Medicine and Public Health, University of Wisconsin-Madison, Madison, WI, USA. [3] School of Pharmacy, University of Wisconsin-Madison, Madison, WI, USA. [4]These authors contributed equally: Zihui Li, Daniel M. Tremmel. [5]These authors jointly supervised this work: Jon S. Odorico, Lingjun Li. ✉email: jon@surgery.wisc.edu; lingjun.li@wisc.edu

The extracellular matrix (ECM) is the network of proteins and polysaccharides surrounding cells, forming a niche in which cells reside and function. Every tissue has a unique composition and arrangement of ECM components, and ECM is known to be significantly remodeled throughout development, with aging, and during various disease states[1–3]. During development, the ECM changes in both composition and organization, guiding cell migration and influencing cell fate[1]. In cell physiology, ECM plays a multitude of roles, from providing structural and mechanical support to modifying growth factor diffusion and binding and providing anchorage-dependent cell survival signals[4,5]. Modern approaches in regenerative medicine and tissue engineering aim to utilize or recapitulate ECM scaffolding and signaling[6,7], but only recently have studies been able to extensively define the matrisome of human tissues[8], and studies of ECM composition throughout human development are even more limited. Understanding the complexity of ECM in normal tissue, throughout distinct developmental time points, can provide important context for understanding healthy physiology, identifying and characterizing disease states, improving cell and tissue isolation methods, and recapitulating physiological tissues or development in vitro.

The pancreas is a glandular organ which functions in both digestive and endocrine systems. The exocrine cells in the acini secrete digestive enzymes into a ductal system eventually emptying into the duodenum. The islets of Langerhans are clusters of five endocrine cell types in close contact with capillary networks, which secrete hormones into the blood. Pancreatic ECM has been studied in many non-human species and across developmental time frames, and has been found to have significant compositional and structural differences among species and from fetal to adult tissues[9]. The ECM of the pancreas is also known to be heavily altered in states of fibrotic pancreatitis and with the progression of pancreatic cancers[10–12]. Islets are surrounded by a capsule consisting of fibroblasts and ECM[13,14]; along the vasculature that penetrates the islets, there is a double basement membrane between endocrine cells and islet capillaries, which is unique to human islets[15]. Furthermore, islet and capillary ECM is altered in the progression of both type 1 and type 2 diabetes[16–19], and found to play important roles in the survival and function of insulin-secreting beta cells[9,18,20]. The role of the ECM in the pancreas, and specifically within islets, is of key interest in the fields of diabetes and beta cell replacement.

To better understand the usefulness of ECM for regenerative medicine and tissue engineering applications, it is essential to have a more complete and comprehensive understanding of the ECM within normal, native tissue. Previous data about the pancreatic matrisome are abundant, but often incomplete and inconsistent. Ample data have been derived from various animal species, but variation among species is well documented[14,21] and even the architecture of the islets themselves differs dramatically between rodents and primates[15,22–27]. Until recently, most data on human pancreas and islet ECM have been obtained through immunohistological staining, and thus have not comprehensively identified the compositional profile of ECM proteins. As mass spectrometry (MS) becomes an essential tool for detecting biomolecules, MS-based proteomics shows great potential for the sensitive and large-scale analysis of complex biological systems. Characterization of ECM proteins using MS, however, still faces challenges due to their unique biophysical and biochemical properties. The large dynamic range of the analytes renders it relatively intractable to in-depth analysis, and heavy crosslinking between ECM and other components results in low solubility and therefore decreased identification rates[28]. To overcome these difficulties, various orthogonal separation techniques were utilized prior to liquid chromatography with tandem mass spectrometry (LC-MS/MS) to decrease sample complexity including off-gel electrophoresis[29], basic reversed-phase LC[30], and strong cation exchange (SCX)[31]. Different protein extraction, ECM enrichment, and digestion methods have also been explored to improve ECM coverage[32–38]. As most of these approaches require extensive sample extraction or fractionation, the higher cost of additional sample handling and lengthy LC-MS/MS runs should be considered and balanced against improved proteome coverage[39].

To compare the relative abundance of ECM proteins from different samples, label-free quantification, which is based on measuring precursor ion intensities or spectral counting, has been broadly used in related studies[8,40]. Although easy to perform, label-free quantification methods demand longer instrument time and the quantification is less accurate due to run-to-run variation[41]. Label-based approaches, on the other hand, allow for accurate and multiplexing quantitative analysis. Among these labeling methods, commercially available isobaric tags such as tandem mass tag (TMT)[11,42] or isobaric tags for relative and absolute quantitation (iTRAQ)[43–45] have been more widely used recently. Nevertheless, the utility of these commercial isobaric tags in large-scale, discovery proteomics studies is often hampered by the limited multiplexing capacity and the high price of the reagent kits. To address these limitations, we have developed a cost-effective alternative, based on a set of $N,N$-dimethyl leucine (DiLeu) isobaric tags that can be synthesized in-house at high yield, with three steps using commercially available reagents, at a fraction of the cost. DiLeu isobaric tags were initially demonstrated as a 4-plex set[46], and then expanded to an 8-plex set[47]. The multiplexing capacity was then increased threefold from 4-plex to 12-plex by taking advantage of the mass-defect feature of stable isotopes to enable simultaneous quantification of 12 samples via 12 reporter ions spanning from 115 to 118 $m/z$[48].

Utilizing the aforementioned methods, a few studies have investigated ECM proteins in isolated mouse islets[45], decellularized rat pancreatic tissue[49], and human pancreas with pancreatic ductal adenocarcinoma and pancreatitis[11]. In addition, we recently reported a MS-proteomic ECM analysis comparing human fetal and adult pancreas using a sample preparation protocol known as the surfactant and chaotropic agent-assisted sequential extraction/on-pellet digestion (SCAD)[31,50]. However, an in-depth quantitative study of pancreatic ECM throughout human development and maturation has not yet been reported.

Herein, we present a relatively simple and effective workflow for quantitative ECM analysis with our custom-developed 12-plex DiLeu isobaric tags and apply this pipeline to investigating matrisome alterations at various developmental stages of human pancreatic tissue. These results add to the knowledge base in understanding the significant alterations that occur in the ECM and associated extracellular molecules throughout the human life cycle. They also provide valuable information for future regenerative medicine strategies for beta cell replacement.

## Results

**In-depth proteome-wide and ECM quantification using 12-plex DiLeu isobaric tagging.** We performed quantitative analysis on normal, non-diabetic, non-pancreatitis human pancreatic tissue from four developmental stages, including fetal (18–20 weeks gestation), juvenile (5–16 years), young adults (21–29 years), and older adults (50–61 years) (Fig. 1) (donor information is available in Supplementary Data 1). The SCAD method, which takes advantage of the solubilizing power of both surfactant (sodium dodecyl sulfate (SDS)) and chaotropic (urea) reagents, together with on-pellet digestion, renders a more comprehensive protein extraction and improves digestion efficiency and recovery of

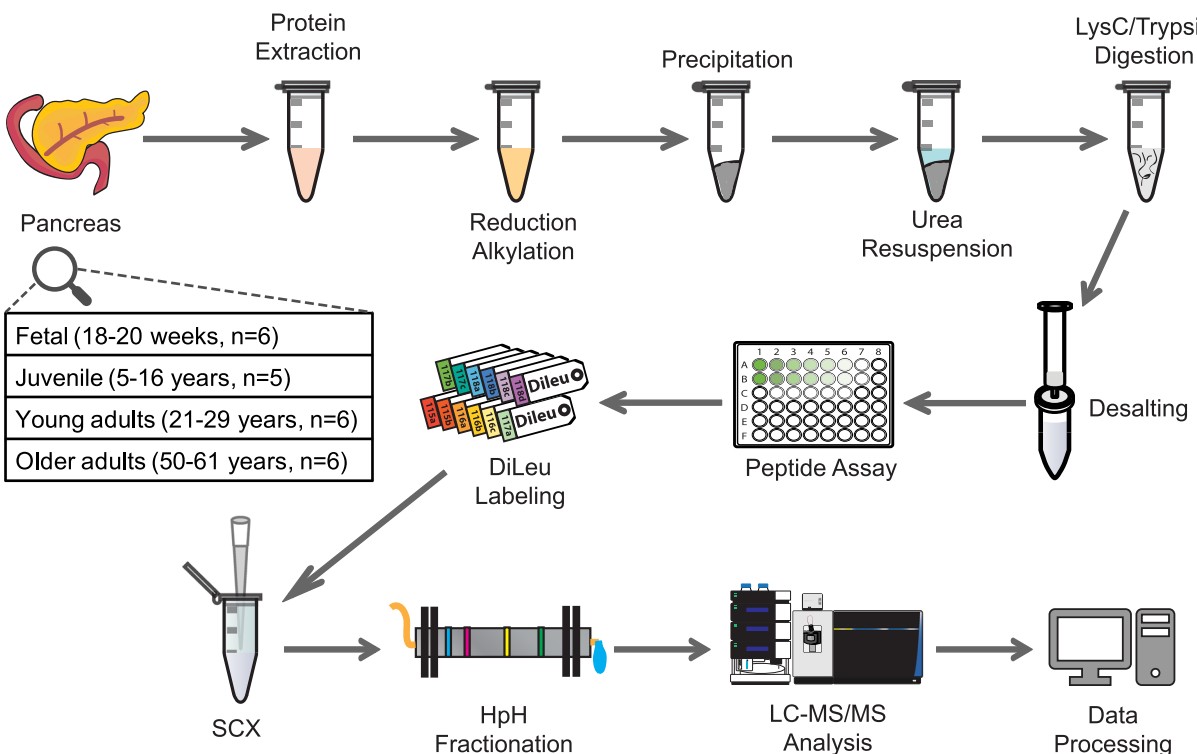

**Fig. 1 Experimental workflow of quantitative analysis using 12-plex DiLeu isobaric labeling strategy.** Pancreata from four age groups were subjected to protein extraction with SDS buffer, precipitation, and digestion according to a slightly modified SCAD method. 12-plex DiLeu isobaric labeling was used to achieve quantitative proteomics analysis. Sample clean-up and fractionation steps were performed prior to LC-MS/MS analysis and data processing was done with commercially available software packages. SCX strong cation exchange, HpH high pH.

relatively insoluble pellets[31]. We used two sets of 12-plex DiLeu tags to label these samples, while a shared sample was involved in each set to allow for better normalization and quantitation. We also replaced the previously utilized SCX[31] with off-line high pH (HpH) fractionation prior to LC-MS/MS which reduced sample complexity, thus not only improving proteome coverage but also strengthening detection of low-abundance proteins[51]. In total, we identified 3523 proteins with high confidence and quantified 2064 that were present across all samples (Supplementary Data 2 and 3). Among them, 185 proteins were categorized as ECM proteins based on Human Matrisome Database[29,52,53] and 117 were quantifiable (Supplementary Data 4 and 5), which makes it one of the largest datasets of human pancreas matrisome. The identified proteins show a large dynamic range spanning over six orders of magnitude (Supplementary Fig. 1) that demonstrates the ability of our method to detect low-abundance molecules.

**Proteome-wide alterations at different developmental stages of human pancreas.** We performed hierarchical clustering of all quantified proteins to explore their profiles at different stages (Fig. 2a). This heat map illustrates column-wise clustering of biological replicates in either the fetal or juvenile group, suggesting larger intergroup differences than intragroup variations. Samples from young adults and older adults, on the other hand, are mixed and grouped together, which suggests that the biological environment becomes relatively stable in adulthood. Our workflow shows the capability of accurate protein quantification and our samples are representative of a range of age groups to produce conclusive results. To examine the reproducibility of samples within each group more carefully, we performed pairwise Pearson's correlation analysis and the

results were summarized as a heat map (Supplementary Fig. 2a). Intragroup samples exhibit good correlation with one another, with an average coefficient over 0.9, while shallow-color regions show that samples from different groups are more distinct. We did not observe obvious gender disparities according to dendrograms generated from hierarchical clustering, but a confirmatory conclusion might need verification with larger sample cohorts (Supplementary Fig. 2b). We then conducted one-way analysis of variance (ANOVA) to compare protein abundance across multiple stages and found that 1570 proteins were significantly changed (false discovery rate (FDR) 0.05) (Supplementary Data 3). The profiles of these proteins are depicted by hierarchical clustering (Fig. 2b). Six clusters can be further generated based on their changing patterns across the four developmental stages and selected biological processes and pathways are annotated for each cluster. For better pairwise comparison between groups, we performed Student's *t* test of all combinations and graphically displayed the results using volcano plots (Fig. 2c). Colored dots refer to significantly changed proteins (*p* value < 0.05) identified with a fold change > 2. We examined more closely two pairs of samples: juvenile versus fetal, which reflects postnatal maturation, and young adults versus juvenile, which reflects postpubertal maturation of the pancreas. Gene set variation analysis (GSVA)[54,55] was performed to reveal expression changes of functionally related genes or gene sets. Heat maps show the significantly changed cellular components and molecular functions, while the color coding indicates normalized enrichment scores in each sample (Supplementary Fig. 3). Specifically, we found some terms related to the exocrine function of pancreas, such as lipase and exopeptidase activity, were highly expressed in older age groups, which is a clear sign of organ maturation. A full list of

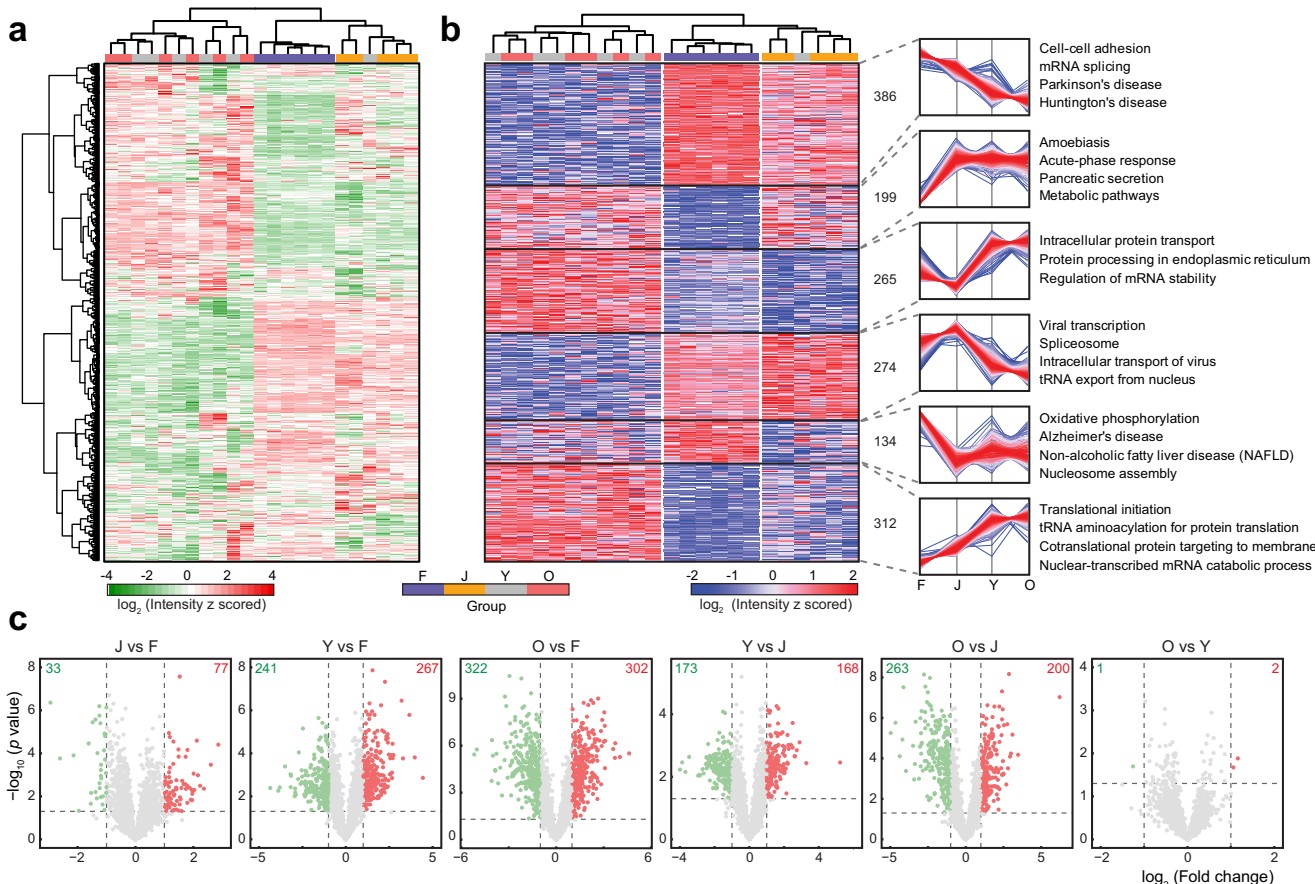

**Fig. 2 Protein profiles in pancreatic tissue and alterations across multiple developmental stages. a** Hierarchical clustering of DiLeu reporter ion intensities of 2064 quantified proteins. **b** Hierarchical clustering of DiLeu reporter ion intensities of 1570 significantly changed proteins (one-way ANOVA, FDR 0.05). Six clusters of these proteins are depicted based on different intensity profiles across four developmental stages. Color of each line in clusters is based on its distance from the center, red (close) to blue (far). The number of proteins and selected enriched biological processes and pathways are indicated for each cluster. **c** Volcano plots showing pairwise comparisons of protein expression levels between various stages. Points above horizontal dashed lines represent significantly altered proteins (two-sided $t$ test, $p$ value < 0.05, $p$ values were adjusted by Benjamini–Hochberg correction for multiple comparisons). Significantly downregulated proteins (e.g., lower abundance in J in the first plot) are shown in green (protein fold change < 0.5) and upregulated ones are shown in red (protein fold change > 2). F fetal, J juvenile, Y young adult, O older adult. Source data are provided as a Source Data file.

enriched cellular components, molecular functions, biological processes, and transcription factor targets in GSVA is also provided (Supplementary Data 6). Biological processes enriched from significantly changed proteins also indicate highly distinct molecular features between each pair (Supplementary Fig. 4a, c). Each node refers to an enriched term and different terms are grouped into clusters based on their similarities, while the most statistically significant term represents the cluster name[56]. We also generated a chord diagram[57] for each pair to further discern how proteins change within a few processes related to pancreas functions, including response to glucose and response to peptide and regulated exocytosis (Supplementary Fig. 4b, d). Results indicate the complexity in the regulation of these biological processes with slightly more upregulated proteins involved. We also found that many previously reported pancreatic cancer biomarkers showed different expression levels at various developmental stages (Supplementary Fig. 5). These findings, along with other interesting correlations not presented, warrant further investigation.

**ECM remodeling of the human pancreas throughout life.** Hierarchical clustering of all quantified ECM proteins suggested

matrisome features unique to different age groups, including fetal, juvenile, and adult (Fig. 3a). Pairwise comparisons showed significant ECM compositional changes throughout the fetal, juvenile, and adult stages, although between young and older adults very few significant differences in ECM were found (Fig. 3b). ANOVA analysis revealed that 84 ECM proteins were significantly changed accounting for 72% of the 117 quantified ECM proteins (Supplementary Data 5 and Supplementary Fig. 6a). Categorical differences of the significantly changed ECM proteins between age groups were also observed. While only 7 of 19 (37%) collagens that were quantified changed in abundance among the developmental groups, the majority of ECM glycoproteins (26/35, 74%), ECM regulators (25/28, 89%), and ECM-affiliated proteins (18/20, 90%) were found to change (Supplementary Fig. 6b, c). The expression levels across developmental stages were compared to select individual ECM proteins that exhibited high abundance in one or more developmental groups, or that showed significant expression changes (Fig. 3c and Supplementary Fig. 7). This analysis highlights at least four patterns of changing expression levels. A subset of collagens shows similar abundance across all stages (e.g., COL1A1, COL3A1, COL5A1, COL6A1). In contrast, a few proteins are expressed at the highest level in the fetal samples and steadily decrease in the juvenile and adult

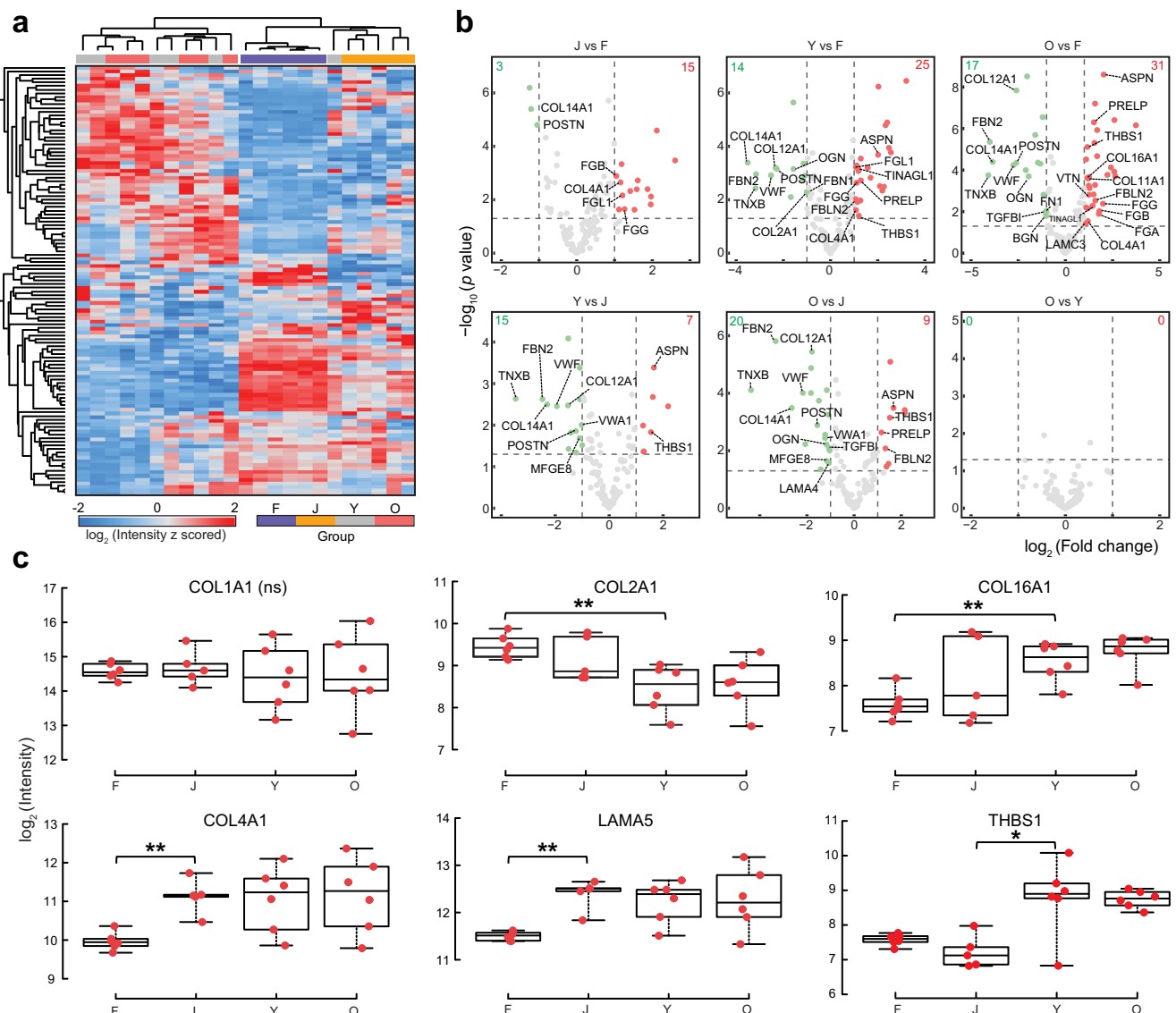

**Fig. 3 ECM remodeling during fetal and postnatal stages of human pancreas development. a** Hierarchical clustering of DiLeu reporter ion intensities of 117 quantified ECM proteins. **b** Volcano plots showing pairwise comparisons of ECM protein expression levels between various stages. Points above horizontal dashed lines represent significantly altered proteins (two-sided t test, p value < 0.05, p values were adjusted by Benjamini–Hochberg correction for multiple comparisons). Significantly downregulated proteins are shown in green (protein fold change < 0.5) and upregulated ones are shown in red (protein fold change > 2). Core matrisome proteins are annotated in each figure. **c** Box plots showing expression levels of selected proteins at different developmental stages. Red dots indicate replicate data points (F, Y, O: N = 6 donors per group; J: N = 5 donors). All box plots indicate median (center line), 25th and 75th percentiles (bounds of box), and minimum and maximum (whiskers). Significance level is marked with an asterisk (two-sided t test, p values were adjusted by Benjamini–Hochberg correction for multiple comparisons, *p value < 0.05, **p value < 0.01; n.s., not significant). F fetal, J juvenile, Y young adult, O older adult. Source data are provided as a Source Data file; exact p values are provided in the Source Data file.

developmental transitions (e.g., COL12A1, COL14A1, FBN2, POSTN, OGN). Others maintain an equal high expression level in the fetal and juvenile stages and only start to decrease at the adult stages (e.g., COL2A1, LAMA4, EMILIN1, FN1). The final group exhibits a lower level in fetal and higher levels in postnatal pancreata (e.g., COL4A1, COL16A1, LAMA5). For four collagen proteins (COL1, COL4, COL5, COL6), multiple isoforms were detected. COL1, COL4, and COL6 have relatively consistent ratios of the measured isoforms throughout all developmental stages studied. Interestingly, the relative ratio of COL5 isoforms changed; COL5A1 and COL5A2 slightly decreased with age, and COL5A3 became more abundant in the adult groups (Supplementary Fig. 7b).

**Visualizing ECM proteins throughout human pancreas development.** To confirm the trends in core ECM protein expression from the MS study (Supplementary Data 7), we performed immunofluorescent staining to visualize a select group of 16 ECM proteins. Representative images are shown in Fig. 4, additional images are included in Supplementary Fig. 8, and higher magnification images in Supplementary Fig. 9 for better visualization of subcellular localization. Proteins found to have nonsignificant differences in total expression by MS among the four developmental groups (COL1A1, COL3A1, COL5A1, COL6A1) were found to also qualitatively exhibit consistent expression by immunofluorescence across developmental ages. Likewise, proteins expressed at higher relative levels during the fetal stage than

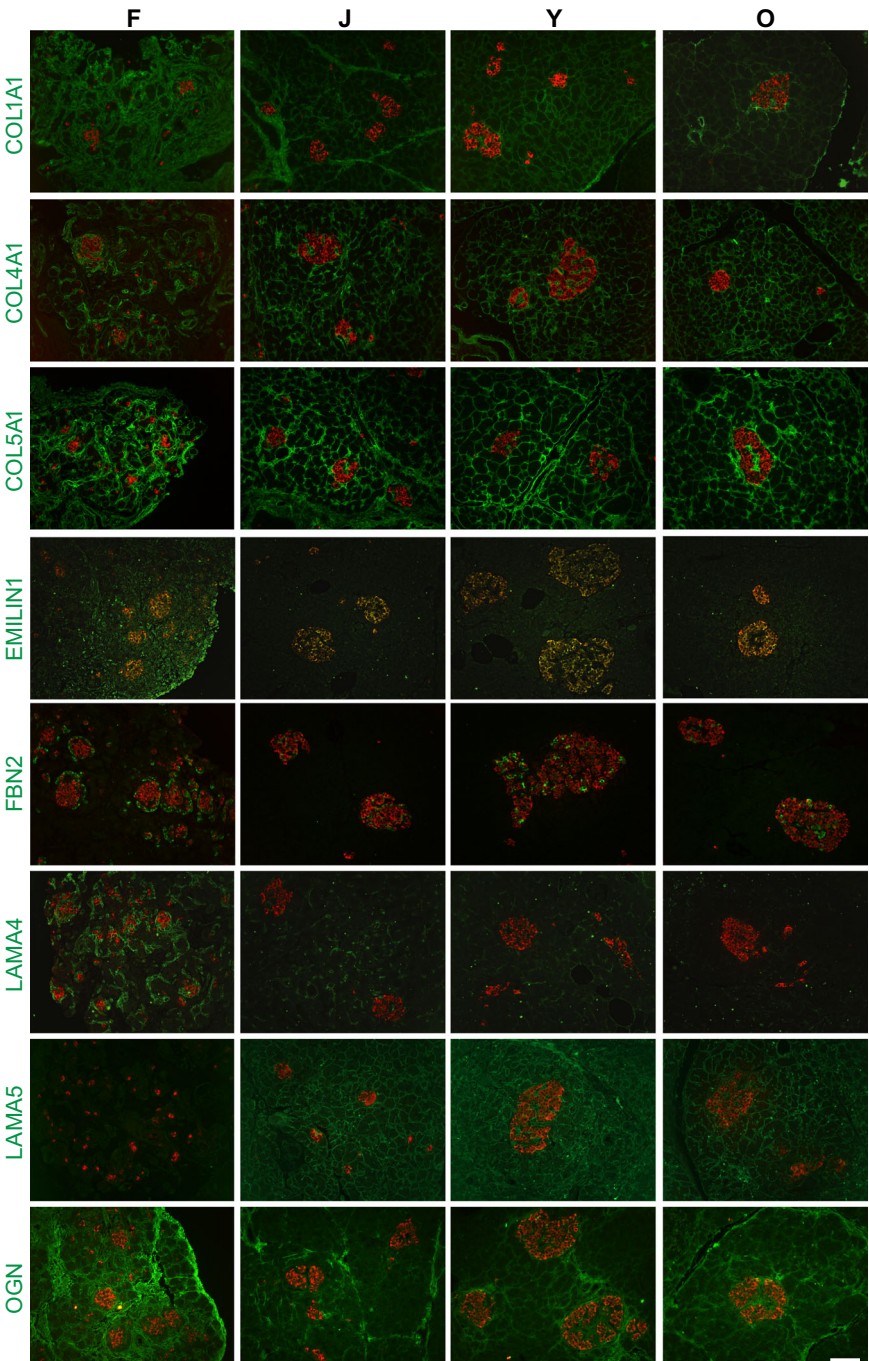

**Fig. 4 Visualizing ECM proteins across multiple developmental stages.** Immunofluorescent images of selected ECM proteins (green) co-stained with insulin (red) in fetal (F), juvenile (J), young adult (Y), and older adult (O) pancreata. Qualitative trends in protein levels corroborate MS data. Representative images are shown, and images were taken for $N = 3$ donors per developmental group. An expanded panel of images is available in Supplementary Figs. 8 and 9. Scale bar $= 100 \, \mu m$.

postnatal stages (COL2A1, COL12A1, COL14A1, EMILIN1, FBN2, FN1, LAMA4, OGN, POSTN) and proteins expressed at lower relative levels in the fetal stage compared to expression at later stages (COL4A1, COL16A1, LAMA5) consistently displayed similar patterns by immunofluorescence staining as they did by MS. Qualitatively, the immunofluorescence staining correlates well with and validates the matrisome data.

**ECM abundance and localization changes across developmental time points.** The visualization of these proteins within

tissue sections provides an opportunity for more precise characterization of ECM localization among the various compartments of the pancreatic tissue. Of specific interest, we quantified the expression of these proteins in the acinar regions, which make up the majority of the pancreas volume, and the islet regions, which constitute ~1–2% of the pancreas. The ratio of signal in the islet and acinar compartments (islet/acinar) was calculated for each image. This ratio represents the enrichment of each particular ECM protein in islets compared to the acinar for each image, but due to the normalization does not account for the total abundance of the ECM protein itself (Fig. 5a, purple bars). Total

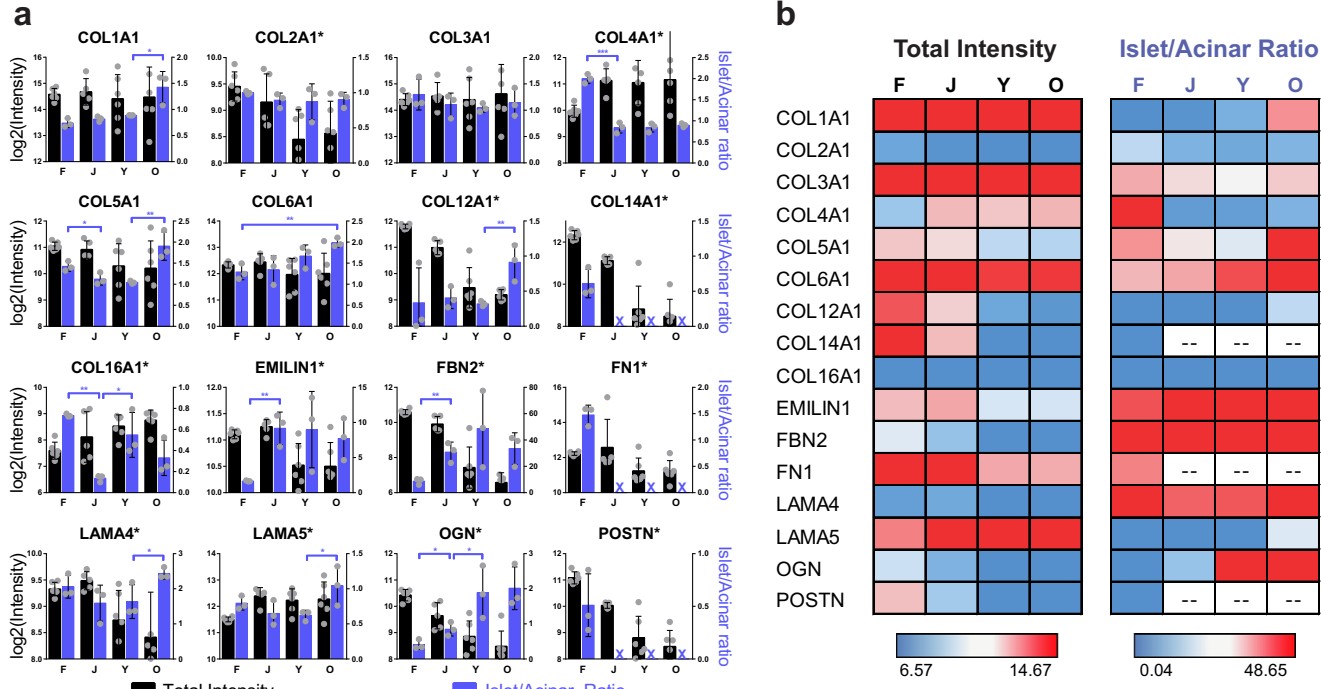

**Fig. 5 ECM proteins change in abundance and localization across developmental time points. a** Representation of total intensity of selected ECM proteins as measured by MS in the whole pancreas (black bars) (F, Y, O: $N = 6$ donors per group) (J: $N = 5$ donors). Quantification of immunofluorescence (IF) staining in pancreatic acinar and islet regions, depicted as the ratio of islet/acinar (purple bars), in fetal (F), juvenile (J), young adult (Y), and older adult (O) pancreata, graphed as mean ± standard deviation ($N = 3$ donors per group). Significant difference for islet/acinar ratio was determined by a two-tailed $t$ test (*$p < 0.05$, **$p < 0.01$, and ***$p < 0.001$). An asterisk next to the protein name indicates a significant difference in total protein abundance by ANOVA among the four groups (FDR 0.05). A purple "X" indicates IF staining was too low in the islets and acinar to calculate the islet/acinar ratio. Source data are provided as a Source Data file; exact $p$ values are included in the Source Data file. **b** Heat maps depicting the same data as in panel (**a**), with emphasis on relative changes throughout the four groups. Blue color indicates the lower 25th percentile of values, red color indicates the top 75th percentile of values, with a gradient toward white at the 50th percentile. Exact values for each element of the heat map are included in the Source Data file.

abundance as determined by MS is included for side-by-side comparison (Fig. 5a, black bars). Heat maps are included (Fig. 5b) to express overall trends in the datasets.

For many ECM proteins, both abundance and localization were found to change throughout development. Only two of the selected proteins (COL2A1, COL3A1) were expressed at similar ratios in the acinar and islet compartments at all four stages. A subset of ECM proteins studied is generally enriched within the acinar compartment (COL12A1, COL14A1, COL16A1, LAMA5, POSTN). Two proteins (COL12A1, COL14A1) are abundant in fetal acinar tissue, whereas COL16A1 is more strongly expressed within the postnatal acinar compartment. Other ECM proteins are enriched in and around the islets (EMILIN1, FBN2, LAMA4, OGN). Both OGN and EMILIN1 are expressed throughout fetal tissue, but become restricted to the islets in postnatal pancreata; EMILIN1 is concentrated intracellularly in the islets, while OGN is localized to the islet extracellular space and, to a lesser extent, in the nearby acini. FBN2 is intracellularly expressed in delta cells, but not alpha or beta cells (Fig. 4 and Supplementary Fig. 11b). COL4A1 and LAMA4 are both enriched in fetal islets with an islet/acinar ratio of ~2. COL4A1 and LAMA4 proteins are localized around the blood vessels of developing islets; however, beyond the fetal stage, COL4A1 is found to be expressed more evenly throughout acinar regions. LAMA4, on the other hand, becomes more enriched within islets in older adults, as do three other ECM proteins in our study (COL1A1, COL5A1, LAMA5). Although we found no significant changes in the total abundance of those proteins between younger and older adult pancreata, there does appear to be significant accumulation in and around

the islets of the older donors. Three proteins (COL14A1, FN1 and POSTN) displayed detectable protein expression in all four groups revealed by MS, but quantifiable staining in islets and acinar was only detectable at the fetal stage. For COL14A1 and POSTN, most expression was found in the fetal mesenchyme (Supplementary Fig. 8). In the adult donors, positive staining for both COL14A1 and POSTN was concentrated around blood vessels and ducts (Supplementary Fig. 11a), with minimal extracellular expression in the islet or acinar compartments, although COL14A1 had faint intracellular staining in the acinar regions. FN1 was found extracellularly in and around the islets in the fetal tissue, but in postnatal tissues was only found faintly and intracellularly in the islets and acinar regions, compared to more robust signal around ducts and vessels, as well as sporadic extracellular staining in the parenchyma, particularly along the boundaries of pancreatic lobules.

## Discussion

Although dynamic proteome changes are known to occur with aging[2,3,58], a comprehensive analysis of the proteome and matrisome across early and late developmental stages of human pancreatic tissue has not been reported. In this study, we present a simple and cost-efficient protocol to achieve in-depth proteome-wide and ECM-specific quantitative analysis which omits labor-intensive extraction or enrichment steps. We utilized this method to systematically examine the dynamics of proteomic composition in human pancreata among four developmental age groups, including fetal, juvenile, young adults, and older adults. We quantified 2064 proteins and found 1570 of them were

significantly changed across multiple stages that suggests a large degree of proteome remodeling through the human life cycle. The identified yet unquantifiable proteins are the result of differences between two batches of DiLeu labeling, which could reflect the distinct protein profiles between different age groups. In addition, the stochastic nature of data-dependent acquisition sampling may also lead to some proteins being below the limit of detection and categorized as undetected across multiple batches[59]. To alleviate this issue, one may consider alternative highly multiplexed labeling strategies such as cPILOT[60] or 27-plex TMT[61]. However, these methods are associated with increased spectra complexity and decreased identification rates, which can be emerging concerns. It is also worthwhile to note that we only collected four fractions and ran two technical replicates for each sample but still achieved a relatively deep analysis. Protein coverage can be further improved by simply increasing the number of fractions if instrument resources permit. Processes or pathways related to the functions of the pancreas are mainly activated with increasing age such as pancreatic secretion and metabolic pathways. On the other hand, some cancer-related pathways are more often deactivated with age, which is likely a sign of postnatal maturation of the organ. We observed many significant compositional changes from fetal to juvenile and juvenile to young adult pancreata. Young and older adults, on the other hand, are more similar in composition, with very few significantly altered proteins. We examined some interesting targets more closely. As an example, thrombospondin 1 is a glycoprotein that mediates cell-to-cell and cell-to-matrix interactions, which is therefore involved in tissue genesis and remodeling[62]. In our study, it showed a >2-fold increase in abundance from juvenile to young adult that is consistent with organ maturation. In addition, thrombospondin 1 has been shown to be expressed at high levels during tumor progression[62,63] and numerous thrombospondin-based therapeutic approaches have been studied[64]. However, our data suggest that conclusions or strategies based on these studies could be further strengthened by taking into consideration the age of the patient. Along this direction, we found that many previously reported pancreatic cancer biomarkers showed different expression levels at various developmental stages such as ANXA2[65], PLG[65], ITGB1[65], ACTN4[66], LGALS1[66], and LAMB1[66]. ACTN4 and LGALS1, for instance, were reported to show fold changes of 1.85 and 1.75, respectively, in cancerous versus normal tissue. Nevertheless, they also have >1.3-fold higher expression in young adults compared to juvenile and this difference is even more dramatic in older adults. Although few studies mentioned an age-dependent effect on biomarker protein expression, our results indicate age-related alterations and the necessity of paying particular attention to patient age in biomarker discovery or disease treatment studies.

In regard to pancreas and islet ECM, previous studies mainly focused on a subset of collagens or laminins, using techniques to quantify specific ECM proteins through gene expression and immunofluorescent staining[15,67]. In our study, we have identified 185 ECM proteins and quantified 117 throughout four developmental stages of the human pancreas. Eighty-four of those quantified ECM proteins were significantly changed in total abundance among the four age groups and displayed distinct expression patterns, which revealed the complexity of ECM remodeling during development and maturation. Very few previous studies reported on the changes in ECM abundance or localization among human fetal and adult pancreas tissue. Otonkoski et al.[15] observed the distribution of laminins throughout the acinar and islet regions of fetal and adult human pancreas by immunofluorescent staining. The authors found that the profiles were quite similar between the two age groups, with the major differences being that LAMA1 was expressed in fetal

but not in adult islets, and LAMB2 was only expressed in adult islets. Likewise, we found that LAMB2 was expressed at significantly lower levels in the fetal tissue than in adult tissue, but not absent, while LAMA1 was only detected in fetal and juvenile tissues (Supplementary Data 4). Extending these findings to other laminins, our results indicate a general shift in the composition of laminins between fetal and adult stages, with some laminins increasing in abundance from fetal to adult (LAMA2, LAMA5, LAMB2, LAMC3) and only LAMA4 decreasing. LAMB1 and LAMC1 did not significantly change throughout the developmental stages we measured. We further identified ECM proteins only present in some developmental groups, which could reflect the most drastic differences between developmental stages, and these proteins were not quantifiable due to only being detected in a subset of the donors. Fifty-nine ECM-related proteins were exclusively detected in the fetal and juvenile groups and nine were found only in adult samples (Supplementary Data 4). For instance, FMOD, which has critical roles in the ECM organization as well as the initiation and progression of several malignancies[68], is present exclusively in the fetal and juvenile groups. On the other hand, MXRA5 that functions in an anti-inflammatory and anti-fibrotic role[69] is only detected in the adult samples. These observations can provide deeper insights into ECM dynamics during pancreas development. Overall, we have found that as a group, collagens change less dramatically throughout development, while glycoproteins and proteoglycans, which contain ECM proteins that contribute to the basement membrane (e.g., laminins, perlecan), exhibit much more dynamic changing patterns throughout development (Supplementary Fig. 6c).

A primary means of studying islet function is to use islets isolated from the pancreas through enzymatic treatment, which destroys much of the native ECM and disrupts islet architecture[70,71]. This fact has prompted many studies to supplement ECM for improved islet culture, and the limitations to the longevity of islet culture highlight the necessity of ECM for islet survival and function[9,20,72,73]. Studies using human fetal isolated islets have also identified a role for ECM in the differentiation, proliferation, and function of islet endocrine cells during in utero development[67,74–76]. As many look to use stem cell-derived islets as a future diabetes therapy, some question the role ECM may have in the differentiation, function, and transplantation of these cells[77–82]. To better understand the islet ECM environment, we investigated the localization of selected ECM proteins in addition to total abundance in the pancreatic tissue.

Four of the ECM proteins we studied were enriched in adult islets relative to the exocrine tissue (COL6A1, EMILIN1, FBN2, OGN). Importantly, the relative abundance and distribution of these proteins was found to also change throughout development. OGN, for example, is expressed throughout the acinar and islet regions of fetal and juvenile pancreata, whereas in adult pancreata it is more concentrated in islets. One previous study has found a role for OGN in islet function; whole-body OGN knockout mice have impaired glucose tolerance and reduced beta cell mass, while OGN supplemented in vitro or at a systemic level in vivo, improves beta cell function and glucose tolerance[83]. While this study focused on circulating OGN produced from osteoblastic cells, our finding that OGN is concentrated around human pancreatic islets suggests that OGN may also play a localized role in islet function. Furthermore, FBN2 has been reported as one of 13 methylated genes silenced in human pancreatic cancers[84], but to our knowledge no other studies have reported on the normal function of FBN2 in the pancreas or in islets. Interestingly, somatostatin was another one of the 13 genes identified in the study[84]; we now additionally show that FBN2 is exclusively expressed within somatostatin-expressing delta cells of the islet.

In addition, a role for EMILIN1 in pancreas or islet physiology has not been reported, but the protein has been identified in previous pancreas and islet matrisome studies[45,85]. Collagen 6 coculture has recently been shown to improve human islet survival and function[86], but it is not clear if collagen 6 is more potent than other purified collagens in enhancing islet in vitro function.

Our present study also revealed unexpected differences in the islet ECM between the younger adult (21–29 years) and older adult (50–61 years) groups. Four ECM proteins (COL1A1, COL5A1, LAMA4, LAMA5) were enriched within the islets in the older adults compared to younger adults, while the total abundance of these proteins in the pancreas was insignificantly different between the two groups. This suggests that the adult islet matrisome changes with age, consistent with recent findings that younger and older adult donors have localized differences in islet ECM, which may impact the effectiveness of enzymatic digestion during islet isolation[87,88].

Research in animal models has indicated that functional changes in islet maturation occur postnatally[89], but the possible correlation with ECM structural changes in human islets has not been studied. A recent study by Arda et al.[90] comparing gene expression and function of isolated human islets from juvenile and adult donors indicated that there are over 500 genes differentially expressed between the two age groups, with distinct histone-mediated changes in regulation of gene expression. Furthermore, the authors observed that age was correlated with functional changes. Adult islets secreted more insulin than juvenile islets under both basal and glucose-stimulated conditions, while total insulin content did not change between juvenile and adult islets. Our study suggests that the islet ECM environment also changes between childhood and adulthood. Further studies into how maturing islets interact with ECM during this key transition are likely to be informative toward better defining the mechanisms of maturation, and risks of autoimmune diabetes.

Likewise, ECM studies using animal tissues and isolated islets have previously been reported, but whether these findings are relevant to human tissue is uncertain. In their 2017 study, Naba et al.[45] identified 120 ECM proteins in isolated mouse islets, including 66 core ECM proteins. The methodology employed in their study does not appear to translate to the larger and denser human pancreas, as effective efforts to isolate islets from the human pancreas inherently destroy the ECM[91]. In the present study, we have identified 61 core ECM proteins in human pancreas across the four developmental stages. Interestingly, many of the significant proteins highlighted in our study were also identified in the Naba study, indicating previously unknown similarities of important ECM components between both species. A total of 80 core ECM proteins are identified between the two studies. Although there were similarities between species, 33 proteins, or 41% of all core ECM proteins, only appear in one of the species' datasets. This difference highlights species-specific characteristics in the pancreatic and islet matrisome that may play relevant roles in health and disease. In a separate unpublished study, all 117 ECM-associated genes that were identified in our proteomic study were also detected at the RNA level in isolated adult human islets, which may drive future efforts to quantify the human islet-specific matrisome and investigate islet-specific changes throughout development.

Overall, we present a quantitative proteomic analysis of human pancreas to delineate molecular changes, particularly ECM composition, throughout pancreas development, maturation, and aging. We found that many previously reported pancreas tumor biomarkers displayed significant changes in protein expression across multiple developmental stages and this age-dependent effect could be important knowledge for studies in biomarker identification and clinical implementation. Our study revealed dynamic changes in pancreatic ECM composition and localization throughout the life cycle, and have identified specific ECM proteins enriched in pancreatic islets (e.g., COL6A1, EMILIN1, FBN2, OGN) which may provide insight for studying islet development, function, and disease. In addition, our protocol can be easily adapted to achieve large-scale and in-depth quantitative analysis of ECM-containing proteome in other biological systems with relatively simple operations and low cost. Our data interpretations are far from exhaustive, but instead, we expect our results to serve as a valuable resource for broad audiences with various interests and will provide a foundation for more in-depth investigations.

## Methods

**Human pancreas tissue preparation**. Human fetal pancreas tissue was obtained from secondary sources (Advanced Biosystems Resources, Inc.) under approved Material Transfer Agreements and with protocols approved by the University of Wisconsin's Institutional Animal Care and Use Committee and Institutional Review Board (IRB) (IRB Study #2013-141). ABR, Inc. obtained consent in accordance with Uniform Anatomical Gift Act and National Organ Transplant Act guidelines. ABR, Inc. warrants that appropriate consent for tissue donation is obtained and adequate records of such consents are maintained. In addition, these tissues were obtained with local, state, and federal laws and regulations governing the procurement of human tissue. Within 24 h of recovery, the organs were received and cleaned of surrounding connective tissue. Small pieces of tissue were removed and fixed with 4% paraformaldehyde for histology, and the majority of the pancreas was homogenized in sterile water for 3 s. The homogenized tissue was pelleted ($16,100 \times g$, 5 min) and the translucent supernatant was discarded. The pellet was flash frozen and stored at $-80\,°C$ prior to further processing for MS analysis.

Juvenile and adult human pancreas tissue was procured by the University of Wisconsin Organ and Tissue Donation Services from donors with no indication of diabetes or pancreatitis, with consent obtained for research from next of kin and authorization by the University of Wisconsin-Madison Health Sciences Institutional Review Board (IRB granted an exempt from protocol approval for studies on postnatal tissue because research on deceased donors is not considered human subjects research). IRB oversight of the project is not required because it does not involve human subjects as recognized by 45 CFR 46.102(f), which defines a "human subject" as "a living individual about whom an investigator (whether professional or student) conducting research obtains (1) data through intervention or interaction with the individual, or (2) identifiable private information." Following organ harvest, pancreata were allocated for research if deemed unfit for transplantation due to vascular damage during organ recovery, no suitable recipient, and nonideal age or body mass index. The organs were received within 24 h of recovery and trimmed of extra-pancreatic connective tissues, including duodenum, large arteries, and veins. The parenchyma was cut into 1 cm³ cubes and frozen at $-80\,°C$ for future use, some pieces were also immediately fixed with 4% paraformaldehyde for histology. One piece of frozen pancreas per donor was thawed and rinsed with 1× phosphate-buffered saline (PBS) followed by sterile water, and then manually chopped into small pieces. The pieces were immersed in sterile water and homogenized for 3 s, then pelleted ($16,100 \times g$, 5 min). Any floating lipids were removed, and the translucent supernatant was discarded. The pellet was flash frozen and stored at $-80\,°C$. Donor information can be found in Supplementary Data 1.

**Protein extraction and digestion**. A slightly modified SCAD method[50] was used to prepare all pancreas samples. Each sample was dissolved in 150 μL of extraction buffer solution (4% SDS, 50 mM Tris buffer) and sonicated using a probe sonicator (Thermo Fisher Scientific). Protein extracts were reduced with 10 mM dithiothreitol for 30 min at room temperature and alkylated with 50 mM iodoacetamide for another 30 min in dark before quenched with dithiothreitol. Proteins were then precipitated with 80% (v/v) cold acetone ($-20\,°C$) overnight. Samples were centrifuged at $14,000 \times g$ for 15 min, after which supernatant containing SDS (in the extraction buffer) was discarded. Pellets were rinsed with cold acetone again and air-dried at room temperature. Eight moles of urea was added to dissolve the pellets and 50 mM Tris buffer was used to dilute the samples to a urea concentration <1 M. On-pellet digestion was performed with LysC/trypsin (Promega) in a 50:1 ratio (protein:enzyme, w/w) at 37 °C overnight. The digestion was quenched with 1% trifluoroacetic acid and samples were desalted with Sep-Pak C18 cartridges (Waters). Concentrations of peptide mixture were measured by peptide assay (Thermo Fisher Scientific). One hundred micrograms of peptide was aliquoted for each sample, dried in vacuo, and reconstituted in 0.5 M triethylammonium bicarbonate prior to DiLeu labeling.

**12-plex DiLeu labeling**. Synthesis of DiLeu tags and labeling process were performed according to protocols previously described[46,48]. Briefly, L-leucine or isotopic L-leucine and sodium cyanoborohydride or sodium cyanoborodeuteride (2.5× molar excess to leucine) were suspended in $H_2O$ or $D_2O$, and the mixture was

cooled in an ice-water bath. Formaldehyde (CH$_2$O, 37% w/w) or isotopic formaldehyde (CD$_2$O or $^{13}$CH$_2$O, 20% w/w) (2.5× molar excess to leucine) was added dropwise, and the mixture was stirred in an ice-water bath for 30 min. The target product was purified by flash column chromatography (MeOH/DCM) and dried in vacuo. Each isotopologue of reporter 115 and 116 requires $^{18}$O exchange prior to reductive dimethylation. Leucine or isotopic leucine was dissolved in 1 N HCl H$_2$$^{18}$O solution (pH 1) and stirred on a hot plate at 65 °C for 4 h. Following evaporation of HCl from the solution in vacuo, trace amounts of acid were removed with StratoSpheres PL-HCO$_3$ MP resin (Agilent) to obtain $^{18}$O leucine in free base form. The identity and purity of DiLeu tags were confirmed with MS before all experiments. One milligram of each DiLeu tag was dissolved in 100 μL of anhydrous N,N-dimethylformamide and combined with 4-(4,6-dimethoxy-1,3,5-triazin-2-yl)-4-methylmorpholinium tetrafluoroborate and N-methylmorpholine at 0.7× molar ratios. The activation was performed by vortexing the mixture for 45 min at room temperature and supernatant was added to each sample for peptide labeling. After vortexing at room temperature for 2 h, the labeling reaction was quenched by addition of hydroxylamine to a concentration of 0.25%. The samples were then dried in vacuo, combined, and cleaned with SCX SpinTips (PolyLC) according to manufacturer's protocols.

**HpH fractionation.** HpH fractionation was performed on a Waters Alliance e2695 HPLC using a C18 reversed-phase column (2.1 × 150 mm$^2$, 5 μm, 100 Å, PolyLC) operating at 0.2 mL/min. Mobile phase A consisted of 10 mM ammonium formate at pH 10 adjusted with ammonium hydroxide and mobile phase B consisted of 90% acetonitrile (ACN) and 10 mM ammonium format at pH 10. Separation was achieved with a gradient as following: 1% B (0–5 min), 1–40% B (5–50 min), 40–60% B (50–54 min), 60–70% B (54–58 min), and 70–100% B (58–59 min). Fractions were collected every 4 min and nonadjacent fractions were concatenated into four samples before being dried in vacuo for LC-MS/MS analysis.

**LC-MS/MS analysis.** Samples were analyzed on an Orbitrap Fusion Lumos Tribrid mass spectrometer (Thermo Fisher Scientific) coupled to a Dionex UltiMate 3000 UPLC system. Each sample was dissolved in 3% ACN, 0.1% formic acid in water before loaded onto a 75 μm inner diameter homemade microcapillary column that is packed with 15 cm of Bridged Ethylene Hybrid C18 particles (1.7 μm, 130 Å, Waters) and fabricated with an integrated emitter tip. Mobile phase A was composed of water and 0.1% formic acid, while mobile phase B was composed of ACN and 0.1% formic acid. LC separation was achieved across a 100-min gradient elution of 3–30% mobile phase B at a flow rate of 300 nL/min. Survey scans of peptide precursors from 300 to 1500 $m/z$ were performed at a resolving power of 60k (at $m/z$ 200) with an AGC target of $2 \times 10^5$ and maximum injection time of 100 ms. The top 20 precursors were then selected for higher energy collisional dissociation fragmentation with a normalized collision energy of 30, an isolation width of 1.0 Da, a resolving power of 60k, an AGC target of $5 \times 10^4$, a maximum injection time of 118 ms, and a lower mass limit of 110 $m/z$. Precursors were subject to dynamic exclusion for 45 s with a 10 p.p.m. tolerance. Each sample was acquired in technical duplicates.

**Data analysis.** Protein identifications and quantifications were performed using Proteome Discoverer (version 2.1, Thermo Scientific). Raw files were searched against the Uniprot Homo sapiens reviewed database (September 2018) using Sequest HT algorithm with trypsin selected as the enzyme and three missed cleavages allowed. Precursor mass tolerance of 20 p.p.m. and a fragment mass tolerance of 0.02 Da were set for the searching. DiLeu labeling on peptide N termini and lysine residues (+145.12801), and carbamidomethylation of cysteine residues (+57.02146 Da) were chosen as static modifications. Dynamic modifications included oxidation of methionine residues (+15.99492 Da), deamidation of asparagine and glutamine residues (+0.98402 Da), and hydroxylation on proline residues (+15.99492 Da). Search results were filtered to 1% FDR at both peptide and protein levels. Quantitation was performed in Proteome Discoverer with a reporter ion integration tolerance of 10 p.p.m. for the most confident centroid. Protein quantitative ratios were determined using a minimum of one quantified peptide. Reporter ion intensities were normalized through equal total peptide amount. ECM proteins were identified and classified by matching the results to Human Matrisome dataset[29]. Missing intensities were replaced using the "replace missing values from normal distribution" feature in Perseus (version 1.6.0.7)[92] prior to further processing. Two-sample Student's t test with a two-tailed distribution for binary comparison and one-way ANOVA analysis were conducted using Perseus. All p values were further adjusted by Benjamini–Hochberg correction for multiple testing. Bioinformatics analyses including Pearson's correlation analysis, hierarchical clustering, protein intensity profiling, volcano plots, GSVA analysis, chord diagram[57], and box plots were achieved using R packages. Biological process network was generated using Metascape (version 3.5)[56] and exported using Cytoscape (version 3.7.1). For protein intensity profiling, biological processes and pathways were enriched using DAVID bioinformatics resources[93] with an FDR cutoff of 0.05. Four selected terms with lower p values were shown for each cluster and the same term was only shown once where it appeared the most significant.

To compare abundance of different proteins, iBAQ method was used embedded in MaxQuant (version 1.5.2.8). Raw files were searched against the Uniprot Homo sapiens reviewed database (September 2018) with trypsin/P selected as the enzyme

and three missed cleavages allowed. DiLeu labeling on peptide N termini and lysine residues (+145.12801), and carbamidomethylation of cysteine residues (+57.02146 Da) were chosen as fixed modifications. Variable modifications included oxidation of methionine residues (+15.99492 Da), deamidation of asparagine and glutamine residues (+0.98402 Da), and hydroxylation on proline residues (+15.99492 Da). The iBAQ method was enabled and all other parameters were set as default.

**Immunofluorescent staining and quantification.** All donor tissues were fixed with 4% paraformaldehyde and embedded in paraffin. Five-micron sections were cut and deparaffinized with xylene and ethanol. Antigen retrieval was performed for 2.5 h at 80 °C in 10 mM sodium citrate buffer. Following washing with PBS-T (1× PBS/0.05% Triton X-100) and blocking for 35 min (1× PBS/10% BSA) at room temperature, antibody-specific staining was performed following Supplementary Data 8. Coverslips were mounted with Fluoromount (Sigma, #F4680). Images were taken at ×20 magnification using a Zeiss Axiovert 200 M microscope with Axio-Vision version 4.8.2.0, and analyzed using the ImageJ software (ImageJ 1.53c). The color channel representing the ECM protein of interest was converted to a binary image using an autothreshold adjustment. Images were analyzed by tracing the islets (Ins$^+$ regions, red) and a neighboring acinar section of about the same size and location as each islet (Ins$^-$ regions, using nuclear arrangement to define acinar clusters); the ECM protein (green) was quantified for each image and normalized by calculating the ratio of intensity within the islet divided by the acinar compartments (islet/acinar ratio). This method is visually outlined in Supplementary Fig. 10. For every combination of antibodies, images were taken for $N = 3$ donors in each developmental group. One to six islets were quantified per image, and 4–5 images per donor, for a total of 9–22 islets quantified per donor, per stain. Statistical analysis was performed with Prism 6 for Windows (version 6.07) (GraphPad Software, Inc.). Results were reported as mean values across biological replicates ± the standard deviation of the mean. For immunofluorescent staining, statistical comparisons between two groups were determined using two-tailed unpaired Student's t tests. A p value <0.05 was considered significant, and Prism's recommended classification for significance was followed ($p < 0.0001$ = extremely significant (****), $0.0001 < p < 0.001$ = extremely significant (***), $0.001 < p < 0.01$ = very significant (**), and $0.01 < p < 0.05$ = significant (*)).

**Reporting summary.** Further information on research design is available in the Nature Research Reporting Summary linked to this article.

## Data availability
The mass spectrometry proteomics data have been deposited to the ProteomeXchange Consortium[94] via the PRIDE[95] partner repository with the dataset identifier PXD020130. An online web application "Matrisome and proteome database of human pancreas" is also available at https://nc-webapp.herokuapp.com/, which enables custom searching of all quantified proteins and supports download of all proteomics datasets in this study. Full immunofluorescence staining results are available from the authors upon reasonable requests. Homo sapiens database used for proteomics data searching was downloaded from Uniprot [https://www.uniprot.org/]. Human matrisome dataset used for ECM protein matching was downloaded from the Matrisome Project [http://matrisomeproject.mit.edu/]. Source data are provided with this paper.

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

## Acknowledgements

This study was supported, in part, by grant funding from the NIH (R21AI126419, R01DK071801, RF1AG052324, P41GM108538, and 1F31DK125021-01), and Juvenile Diabetes Research Foundation (1-PNF-2016-250-S-B and SRA-2016-168-S-B). Data presented here were also in part obtained through support from an NIH/NCATS UL1TR002373 award through the University of Wisconsin Institute for Clinical and Translational Research. The Orbitrap instruments were purchased through the support of an NIH shared instrument grant (NIH-NCRR S10RR029531) and Office of the Vice Chancellor for Research and Graduate Education at the University of Wisconsin-Madison. We would also like to acknowledge the generous support of the University of Wisconsin Organ and Tissue Donation Organization who provided human pancreas for research. Our research team would like to give special thanks to the families who donated tissues for this study. We also acknowledge Sierra Raglin and the University of Wisconsin Department of Surgery Histology Core for help processing and embedding tissues. L.L. acknowledges a Vilas Distinguished Achievement Professorship and the Charles Melbourne Johnson Distinguished Chair Professorship with funding provided by the Wisconsin Alumni Research Foundation and University of Wisconsin-Madison School of Pharmacy.

## Author contributions

Z.L., D.M.T., S.D.S., J.S.O., and L.L. designed the study. Z.L. and D.M.T. performed the experiments and analyzed the data. D.M.T., S.A.M., and A.K.F. processed and banked human tissues. F.M., Y.S., and B.W. were involved in sample preparation and data acquisition. Q.Y. and M.M. were involved in data analysis and interpretation. D.G.D. helped design the website for database searching. D.M.T. and V.S.J. performed immunofluorescent staining and quantification. Z.L., D.M.T., S.D.S., J.S.O., and L.L. prepared the manuscript and all authors provided editorial feedback.

## Competing interests

The authors declare the following competing financial interests: J.S.O. is scientific co-founder of Regenerative Medical Solutions, Inc. He is also Chair of the Scientific Advisory Board and has stock equity. All other co-authors declare no competing interests.
