## [Peer Review File · Nature Communications]

REVIEWER COMMENTS

Reviewer #1 (Remarks to the Author):

This manuscript provides a comprehensive proteomic view of human pancreas of developing stages. The authors included a large number of samples for each stage and the proteomic data has high intragroup reproducibility. The analysis of proteomic data includes pan-proteome and ECM-focused analysis, with a focus on the latter.

The strength of the paper is the high-quality comprehensive human data that will for sure benefit the field. People who come in with their own questions could use the dataset and survey their proteins/pathways of interest. The weakness is that the paper is mainly descriptive without major punchlines that readers can directly take home with. This paper is considered to be a biological paper instead of a methodology paper (MS methodology was published previously), the authors listed many messages, but they are not analyzed to the depth that greatly advances our current understanding of human pancreas development, which dampens my enthusiasm for publication.

Specific comments are given below.

I. Major points:

1. Data analysis on whole proteome is quite limited. For example, GSEA analysis for the pairwise comparisons using the pre-ranked mode with p values – to show the processes/molecular functions/cellular components/transcriptional regulators of the changed proteins during development. Meanwhile, how does this study compare to previous RNA-seq, single cell RNAseq studies in specific stage (for that protein abundance value will be required, see II-3)?

2. ECM is the focus of the analysis. More analysis and discussions about the ECM functional groups, instead of individual proteins, that are changed during development. For example, certain laminins are down and certain are up, a more focus examination on how the basement membrane composition (COLIV, laminins, HSPG2...) changes and a discussion of what was known will be beneficial. When possible, ECM proteins (collagens, laminins) are better considered as molecules with all the chains. Whenever all chains forming one molecule conform in their changes, it improves the power of analysis.

The IF study nicely showed islet vs acinar expression for selected ECM proteins. It would increase to analyze published RNAseq or scRNAseq data in islets to survey the expression of a large set of islet-expressing ECM proteins.

II. Other points:

1. In Fig 2b, no protocol seems to be provided for how to select the enriched biological processes and pathways. FDR and ranking need to be indicated for each process.

2. The identities of the significantly changed proteins, $\log_2(\text{fold change})$, and p value need to be shown as a table (or within Table 3 and 5) for in Fig 2c and Fig3b, respectively, so that readers can sort the data.

3. In Table 3: Is there information about total precursor ion intensity for each protein, which can be split into intensities for that protein in each sample by their $\log_2(\text{intensity})$? The abundance information is missing from the entire analysis. It is important to know the ECM compositions and the more abundant ECM proteins and whether there are over-abundant peptides/proteins that take up most of the MS spectra.

4. In Table 3: Additional information is needed: percent sequence coverage; total number of peptides (i.e., peptide sequence matches) assigned; number of unique peptide sequences; number of spectra.

5. In Fig S9a, some of the artery signal (the yellowish elastic fiber signal) seems to be autofluorescence. A no primary antibody control will help. Scale bars are missing.

6. Zoomed-in images on the ECM IF staining will help readers to decide compartmental (intracellular vs. extracellular) signal.

7. In Fig 3b, what is the criteria for showing the protein labels? Some of the most changed dots are not labeled, such as in JvsF and YvsF panels.

Reviewer #2 (Remarks to the Author):

The manuscript by Li et al. describes the proteomic composition of 23 human pancreas samples across 4 age groups to reveal insight into development and aging associated changes. They employ their diLeu 12plex quantification and SCAD sample preparation methods to obtain very good proteome coverage with a focus on the matric component. Imaging is used to provide context for many of the ECM proteins which provides an excellent resource for regenerative medicine efforts aimed at in vitro and in vivo islet/beta cell therapies. The manuscript is well written, the figures and data also to be of high quality further supporting the importance of this work. There are some limitations to the study that need to be addressed but overall this is a valuable contribution to the field.

Introduction

Line 40 - "but until recently, few studies have been able to extensively define the matrisome of human tissues⁸."

This is not the case there have been over 40 examples to date. Please tone down this claim.

Results

Line 125 – it is mentioned that data spans six orders of magnitude. The figure referred to indicates the iBAQ method was used to estimate protein abundance. This is not described in the methods, please add.

Line 195 – please perform quantitative analysis on the images and report. It is not clear if image analysis was only performed on a representative sample form each age group. Please indicate in the methods at a minimum.

Discussion

Line 356 – "comprehensive". It is not clear that this is a comprehensive analysis (this is very challenging to determine!). consider revising this term. In addition, the sample preparation included a water wash that was discarded. It is almost certain that this fraction contained a significant protein fraction.

Methods

Line 380 – It is an important point that these are not necessarily “normal” tissues collected under ideal conditions.

Line 396 – It is not clear when SDS solution was added. Please add enough detail necessary for others to follow the methods without referring to your previous work.

Line 418 – Earlier in the text it was stated that only 4 fractions were run. Please mention and describe here.

Line 445 – Why were peptides that did not contain all 12 report ions discarded. What is the rationale and describe here if appropriate.

Tables and figures – list the values used for quantification. TABLES 3, 5, 6 All have values that are some log transformed value.

Table 6 - COL1A1 value is 14.57 and COL1A2 is 14.28. these two proteins are usually very close to a 2:1 ratio. Please explain if this is iBAQ log₁₀ and what the corresponding measured ratio is.

Reviewer #3 (Remarks to the Author):

Overall, this is an interesting study conducted by an expert team of investigators.

The authors present proteomic data generated using a custom-developed N,N-dimethyl leucine (DiLeu) isobaric tagging approach in samples of human pancreas of four age groups: fetal, juvenile, young adults and older adults. They identify over 3,500 proteins that change over the course of pancreas development and maturation in these different age groups, including 185 ECM proteins, of which they quantify 117. Some of the ECM proteins were also visualized by immunofluorescence in pancreatic sections.

As a whole, the study adds some new information to other publications on this topic. Data presented are interesting and are likely to inform the research community on future research efforts focusing on the function of individual ECMs in pancreas development.

One of the weaknesses is that the authors make little effort to present their data in a more palatable way for non experts in the field of proteomics. For example, what is the difference between protein signals shown in the vertical columns in panels a and b of Figure 2? What is the actual identity of protein measured? What is the identity of proteins that are significantly changed between age groups shown in the Volcano plots in panel C of Figure 2?

The link provided to access the raw data in the ProteomeXchange Consortium is not particularly helpful for biologists who are non experts in proteomics.

Another limit of the study is that there is no effort to sort out ECM (or other proteins) that are enriched in pancreatic islets. While it is understood that this was done using whole pancreata, the information presented may prove of difficult application to the study of some of these ECMs in islet biology. Some notions about ECMs possibly implicated in islet development and function can only be inferred by a derivative comparison with other published studies (see Naba et al., Scientific reports 7, 40495, 2017).

The expression of some of the ECM identified in the study was validated in situ for only a handful of them, mainly collagens and laminins (which incidentally were already known to be expressed at different levels in different compartments of the pancreas). Osteoglycin (OGN) and periostin (POSTN) are perhaps the only two examples of proteins that have been studied less in the pancreas.

All of the immunolocalization experiments shown, both in Figure 4 and in the Supplemental Figures 7, 8 and 9, lack important negative isotype IgG controls. Without these controls these data have little or no value, especially considering that nowadays companies selling these antibodies provide terrible quality control assurance. Although the authors provide an Excel document where they identify all antibodies used in these immunolocalization experiments, they seem to think that the so-called "Antibody validation information" provided by vendors for each antibody is enough to guarantee the specificity of the antibodies. This is categorically not true, each immunostaining performed needs its own intra-assay IgG control. Historical control experiments conducted on a few tissue sections are not acceptable, as the condition of the tissue sections, the donors, and other variables (e.g. pH of the phosphate buffer used in a given day) may dramatically affect the outcome of the staining.

Most of the Figure legends are extremely succinct, and lack important details for the reader to understand what the data show.

REVIEWER COMMENTS

Reviewer #1 (Remarks to the Author):

This manuscript provides a comprehensive proteomic view of human pancreas of developing stages. The authors included a large number of samples for each stage and the proteomic data has high intragroup reproducibility. The analysis of proteomic data includes pan-proteome and ECM-focused analysis, with a focus on the latter.

The strength of the paper is the high-quality comprehensive human data that will for sure benefit the field. People who come in with their own questions could use the dataset and survey their proteins/pathways of interest. The weakness is that the paper is mainly descriptive without major punchlines that readers can directly take home with. This paper is considered to be a biological paper instead of a methodology paper (MS methodology was published previously), the authors listed many messages, but they are not analyzed to the depth that greatly advances our current understanding of human pancreas development, which dampens my enthusiasm for publication.

Response: We deeply appreciate the keen critique and perspective the reviewer has offered us as guidance to strengthen our manuscript. We indeed hope to provide a resource for the field utilizing this valuable tissue and novel technology, and we are driven to provide this information in an accessible way for other researchers to utilize. While our results are complex, we attempt to present them in a meaningful way, especially regarding the ECM which we are particularly interested in. We thank the reviewer, and have made many changes based on these helpful comments. We hope that we have provided deeper analyses of the results and clarified stronger take-away messages from this work.

Specific comments are given below.

I. Major points:

1. Data analysis on whole proteome is quite limited. For example, GSEA analysis for the pairwise comparisons using the pre-ranked mode with p values – to show the processes/molecular functions/cellular components/transcriptional regulators of the changed proteins during development.

Response: We thank the reviewer for this constructive suggestion. We performed an enrichment of biological processes from significantly changed proteins as in **Supplementary Fig. 4** and pinpointed some processes related to pancreas functions such as regulated exocytosis and response to glucose. Further, we agree that gene set enrichment (GSE) analysis could be a meaningful supplement since it assesses the concerted behavior of functionally related genes forming a set and therefore provides greater sensitivity to find gene expression changes of small magnitude that operate coordinately. We have done a Gene Set Variation Analysis (GSVA) (PMID: 2332383) accordingly, which is a recently improved GSE method that estimates variation of pathway activity over a sample population in an unsupervised manner. We still focused on two pairs of samples of special interest: juvenile versus fetal, which reflects postnatal maturation, and young adults versus juvenile, which reflects post-pubertal maturation of the pancreas. The enriched cellular components and molecular functions (significantly changed between the two age groups, adjusted $p < 0.05$) are summarized as in the heatmaps in **Supplementary Fig. 3**. Color coding in the heatmaps indicates the normalized enrichment score of a certain term in each sample. A full list of significantly changed cellular components, molecular functions, biological processes and transcription factor targets is also provided in **Supplementary Table 6**, which includes the involved genes, p values and fold changes for each enriched term.

We added the following sentences to the “Results” section: “Gene set variation analysis (GSVA) was performed to reveal expression changes of functionally related genes or gene sets. Heatmaps show the significantly changed cellular components and molecular functions while the color coding indicates normalized enrichment scores in each sample (**Supplementary Fig. 3**). Specifically, we found some terms related to the exocrine function of pancreas, such as lipase and exopeptidase activity, were highly expressed

in older age groups, which is a clear indication of organ maturation. A full list of enriched cellular components, molecular functions, biological processes and transcription factor targets in GSVA is also provided (Supplementary Table 6).”

Updated Supplementary Fig. 3:

Supplementary Fig. 3 GSVA analysis in juvenile compared to fetal and young adult compared to juvenile. GSVA analysis showing the significantly changed ($p < 0.05$) cellular components and molecular functions in juvenile versus fetal (a) and young adult versus juvenile (b). Color coding of the heatmaps indicates normalized enrichment score in each sample. A full list of enriched terms including biological processes and transcription factor targets is available in Supplementary Table 6.

Meanwhile, how does this study compare to previous RNA-seq, single cell RNAseq studies in specific stage (for that protein abundance value will be required, see II-3)?

Response: We thank the reviewer for this critical comment, and this is also something we have thought about. Unfortunately, the data of many previous studies were often inconsistent and not very accessible, making it difficult to reuse or compare. Furthermore, our proteomic data is from whole pancreas tissue and focuses on trends over developmental time; it would not be fair to compare our results with many RNAseq or scRNAseq studies which have focused on specific isolated cell types, since the same protein may be expressed in multiple cell types in the pancreas. For these reasons, we did not find a very useful database containing comparable gene expression information for a relevant comparison and this was actually a major motivation of our current study: to establish normal baseline data for future investigations and provide an

easy-to-use database for readers with different scientific backgrounds. Despite all the concerns, we agree it is still worthwhile to perform such a comparison and valuable information could be gained.

For instance, Arda et al. investigated age-dependent islet gene expression in juvenile and adult pancreas (PMID: 27133132). Interestingly, 31 genes that they found to express significantly differently in the two age groups were also identified in our study. Among them, 5 genes showed similar increased expression in adults (SYNPO, FAM3C, HSD17B12, HIST1H1C, VGF) and 13 showed decreased expression level in adults compared to juveniles (HIST1H2AG, PRPH, GSTM2, C4A, HPX, VASP, LMNB1, ANK3, HSPA1B, SELENBP1, FGL1, HSPA1L, LMNA). However, none of these 31 genes corresponded to ECM proteins, which are of special interest in our study. Moreover, Arda et al. did not provide accurate quantitative ratios that would allow for more detailed examinations. While this comparison is still interesting as it confirms that some genes are expressed in an age-dependent manner in both islets and whole pancreas, we feel it is difficult to derive more conclusions from such a comparison. Inconsistencies of the results can easily be explained by methodological differences between the two studies, such as differences in the sample sources, detection techniques (i.e. gene expression vs. protein expression), and sample preparation methods. In addition, the authors did not provide protein abundance information, which would be required to compare the expression of different proteins in comparative age groups between studies.

In another case, Muraro et al. provided a cell-type specific transcriptome atlas of the human pancreas (PMID: 27693023). Among their detected expressed genes, 90 genes in α cells and 175 genes in β cells are also identified as proteins expressed in our study. To compare relative gene expression levels, we ranked these genes using either iBAQ intensities from our study or RNAseq intensities in the literature. The detailed comparisons are shown in the next two pages. There are some similar patterns between the two datasets, for example in α cells, GCG and TUBA1B both rank high as expected. However, inconsistency is also observed, such as annexins ANXA6 and ANXA7 which rank high in iBAQ but lower in RNAseq. Annexins are a group of ECM-affiliated proteins that have been observed to play a role along the exocytotic pathway and they are mostly located inside the cells but also found extracellularly. Again, there are many factors that could lead to a discrepancy of the results between the two studies such as sample differences (i.e. global pancreatic tissue vs. specific cell types), or tissue damage during the islet isolation or sample preparation. Similar studies have also been conducted by Baron et al. (PMID: 27667365) and Wang et al. (PMID: 27364731). However, these two studies did not provide a full list of detected genes or abundance information, which limits possible comparisons or cross-validations.

Overall, the results from many previous studies are often inconsistent or incomplete, and the data are usually difficult to view or mine especially for non-experts. For all these reasons, we didn't include many details of such comparisons in our manuscript. Instead, we tried to make our dataset as complete and accessible as we can by clearly showing all data in supplementary spreadsheets and generating a searchable online database (<https://nc-webapp.herokuapp.com/>) as requested. While bulk RNAseq, microarray or qRT-PCR studies can provide gene expression data of isolated islets, ECM is often damaged by the isolation procedure (PMID: 27456745) and therefore the actual abundance of individual ECM proteins present in isolated islets may be very different from the mRNA abundance determined from gene expression studies. Moreover, impurities of acinar cells in isolated islets may confound compartment-specific gene lists. Single cell RNAseq can provide gene expression information of islet-specific cell types or rare cell populations; however, scRNAseq would not be expected to reveal ECM differences by tissue compartment (i.e. islet vs. acinar) or age-dependent compartmental or global ECM protein expression changes, information which our study uniquely and directly provides.

Comparison of protein abundance ranking in α cells

Gene	iBAQ Rank	RNAseq Rank		iBAQ Rank	RNAseq Rank		iBAQ Rank	RNAseq Rank
GCG	1	1	SDF2L1	31	76	PRKAR1A	61	15
TUBA1B	2	9	CST3	32	43	TMX4	62	53
ALDH1A1	3	4	VKORC1	33	49	ABHD16A	63	67
MYL6	4	11	FLNB	34	57	ACLY	64	29
FKBP2	5	72	CPNE3	35	33	RAB27A	65	63
H1FX	6	80	TSPAN7	36	55	ENPP1	66	85
TMEM205	7	75	PFN2	37	16	GDI1	67	64
COX7A2	8	25	PCBD1	38	38	LCLAT1	68	82
ANXA6	9	68	NDUFB8	39	60	FNDC3A	69	45
ATP5L	10	23	PCSK1N	40	32	GLS	70	24
TUBA1A	11	13	NDUFA7	41	61	CKMT1A	71	83
TMED3	12	58	THEM6	42	86	CHGB	72	3
ECH1	13	54	HSD17B12	43	30	PALLD	73	37
PKM	14	14	COPE	44	81	ZKSCAN1	74	36
FABP5	15	50	CAPZB	45	44	CKB	75	74
NIPSNAP3A	16	78	NDUFB11	46	65	APOA1BP	76	70
CLU	17	6	ESYT1	47	71	ATP6AP2	77	19
KCTD12	18	22	AK2	48	66	MIA2	78	87
ATP5J2	19	40	LUC7L3	49	26	VGF	79	7
ANXA7	20	46	SCG5	50	5	GNAS	80	2
CD63	21	12	INA	51	51	HLA-A	81	8
MYH10	22	41	QDPR	52	31	GC	82	10
SPCS1	23	34	ARHGAP1	53	56	TPD52	83	18
NHP2L1	24	39	SELENBP1	54	90	HMG2	84	28
CNPY2	25	48	PITRM1	55	73	RPS6KA3	85	52
UQCRCQ	26	17	PAPSS2	56	20	IDI1	86	59
NUCB2	27	42	NDUFAF3	57	77	COPZ1	87	62
VAMP2	28	35	HMGB3	58	79	CD36	88	84
MPC2	29	47	KIAA1324	59	21	HIST1H1C	89	88
COA3	30	69	ASPH	60	27	TCEAL5	90	89

Comparison of protein abundance ranking in β cells

Gene	iBAQ Rank	RNAseq Rank		iBAQ Rank	RNAseq Rank		iBAQ Rank	RNAseq Rank
RPL4	1	11	FLOT1	59	87	SNX9	117	110
RPL36AL	2	34	EIF4B	60	21	CDC5L	118	138
RPL5	3	4	ALDOC	61	173	SLK	119	118
RPS23	4	7	G3BP1	62	156	RANBP2	120	83
RPL7A	5	6	VPS35	63	98	HERPUD1	121	76
HSPA8	6	10	TPP1	64	68	WARS	122	162
RPL17	7	30	PURB	65	139	ADRA2A	123	175
HNRNPC	8	36	TMEM33	66	111	WLS	124	147
LMAN1	9	55	DARS	67	88	ATP6V1D	125	109
TMEM109	10	140	ABHD10	68	170	ENDOD1	126	172
RPL3	11	8	ATP6V1A	69	67	TERF2	127	164
RPS4X	12	24	SARS	70	94	GHITM	128	66
HSPD1	13	42	PSIP1	71	152	SLC39A14	129	107
HSP90AB1	14	5	ESD	72	85	PCMT1	130	115
RPS3	15	26	TOMM70A	73	143	LARP1	131	35
CANX	16	23	SEC63	74	82	AP3B1	132	121
PGRMC2	17	77	LRPPRC	75	133	PPL	133	122
ARL6IP5	18	75	PSMB1	76	46	TXNRD1	134	73
HSPA9	19	123	MYH14	77	126	EIF3H	135	108
EEF2	20	13	MAPRE1	78	63	FKBP4	136	116
ENO1	21	28	ATP2A3	79	132	RAD50	137	105
HSPA1B	22	142	ATP6V1B2	80	61	SQSTM1	138	12
RPL31	23	9	FAM213A	81	97	GIGYF2	139	148
HADH	24	32	TOR1AIP1	82	157	STAG2	140	127
YWHAG	25	78	INA	83	91	COPS2	141	145
CLTC	26	27	DNAJC3	84	89	DDX24	142	59
DHRS7	27	71	TMED6	85	171	FASN	143	153
MATR3	28	47	GOLGA4	86	31	TGOLN2	144	40
SEC11C	29	16	FTH1	87	2	GOLGB1	145	52
OCIAD1	30	65	TCEA1	88	70	PRKAR2B	146	174
SURF4	31	22	SRPR	89	50	GTF3C1	147	167
PGK1	32	54	PURA	90	74	SYNE2	148	43
ACAT1	33	159	PKP2	91	168	TRIM2	149	135
CLTA	34	112	NFIC	92	64	MPP6	150	150
HSP90AA1	35	3	EIF4A2	93	18	AARS	151	154
PEBP1	36	14	RAB5A	94	124	TTC37	152	104
PTGES3	37	51	ADH5	95	120	HLTF	153	155
CNP	38	163	KIF5B	96	37	USP9X	154	80
VAMP2	39	58	GNA13	97	160	BAG3	155	161
ST13	40	56	PSMC6	98	137	NEU1	156	113
ANP32B	41	95	PAPSS2	99	29	GNAS	157	1
CISD2	42	128	SCD5	100	44	PSAP	158	20
DSP	43	15	SBDS	101	114	TPD52	159	33
DYNC1H1	44	39	EIF3E	102	101	MEIS2	160	60
SLC25A4	45	84	PRPF8	103	93	PPT1	161	69
ARL8B	46	125	NCKAP1	104	49	PPP2CB	162	72
TSPAN7	47	103	PAFAH1B1	105	90	ENO2	163	81
PFN2	48	25	MYO1D	106	62	TRIP11	164	99
DPYSL2	49	102	ERO1LB	107	19	HUWE1	165	100
CCT4	50	86	TUBA4A	108	134	CDV3	166	106
ALG2	51	144	GFPT1	109	53	SUMO3	167	117
CNBP	52	92	U2SURP	110	131	EIF1B	168	119
ATP6V1E1	53	79	UTRN	111	151	GSN	169	129
RTN4	54	17	TNS1	112	158	PRKACB	170	136
ABAT	55	169	ACLY	113	41	NDRG1	171	141
KTN1	56	48	MIA3	114	130	TXNL1	172	146
TPM3	57	38	EPRS	115	96	BTF3L4	173	149
DBI	58	45	IRF2BP2	116	57	FKBP5	174	165
						HSPA13	175	166

2. ECM is the focus of the analysis. More analysis and discussions about the ECM functional groups, instead of individual proteins, that are changed during development. For example, certain laminins are down and certain are up, a more focus examination on how the basement membrane composition (COLIV, laminins, HSPG2...) changes and a discussion of what was known will be beneficial.

When possible, ECM proteins (collagens, laminins) are better considered as molecules with all the chains. Whenever all chains forming one molecule conform in their changes, it improves the power of analysis.

Response: We thank the reviewer for this helpful suggestion, and we agree for many collagens it will increase the reliability of the results if multiple chains of the same molecule exhibit similar patterns. We looked at several collagens of which we were able to identify all chains, and summarized the results as boxplots in the updated **Supplementary Fig. 7b**. In doing this additional requested analysis, we found that different chains (i.e. alpha 1,2,3) of the same collagen molecule (COL1, 4, 5 and 6) have similar trends at all developmental stages, which improves the confidence of our results. However, we observed some slight deviations, such as changing ratios of the collagen 5 isoforms over developmental time. This distinct expression pattern within the same protein family highlights the unique advantages of using mass spectrometry-based approach to discern much molecular details. In addition, we were not able to identify all chains for some other collagens and this could be attributed to different degrees of sample loss during preparation, or due to distinct ionization efficiency of various proteins in MS. For laminins it is more difficult to make these comparisons since many laminin proteins share the same chains in a partially overlapping fashion. However, to address the reviewer's query, we have added the following sentences in the manuscript: "For four collagen proteins (COL1, COL4, COL5 COL6), multiple isoforms were detected. COL1, COL4, and COL6 have relatively consistent ratios of the measured isoforms throughout all developmental stages studied. Interestingly, the relative ratio of COL5 isoforms changed; COL5A1 and COL5A2 slightly decreased with age, whereas COL5A3 became more abundant in the adult groups (**Supplementary Fig. 7b**)."

We categorized all identified ECM proteins in **Supplementary Fig. 6**, but agree with the reviewer that these categories were not mentioned often in the rest of our analyses. Because we are interested in the extracellular environment of the pancreas, we focused on structural ECM proteins for our subsequent analyses. These proteins are all categorically Collagens, Glycoproteins, or Proteoglycans. We find that as a group, collagen proteins have fewer members that showed significant changes throughout development, while many proteins in the Glycoprotein and Proteoglycan families did change significantly. To emphasize these broader trends, we have added the following sentence into the second paragraph of the "Discussion" section: "Overall, we have found that as a group, collagens change less dramatically throughout development, while glycoproteins and proteoglycans, which contain ECM proteins that contribute to the basement membrane (e.g. laminins, fibronectin, perlecan) exhibit much more dynamic changing patterns throughout development."

Updated Supplementary Fig. 7:

Supplementary Fig. 7 ECM remodeling of human pancreata across developmental stages. Box plots showing expression levels of selected ECM proteins (a) and different chains of the same collagen molecule (b) at different developmental stages. Dots within boxes indicate replicate data points. Top and bottom of boxes indicate 3rd and 1st quartile, respectively, and whiskers extend to maximum and minimum. Horizontal lines within boxes denote the median.

The IF study nicely showed islet vs acinar expression for selected ECM proteins. It would increase to analyze published RNAseq or scRNAseq data in islets to survey the expression of a large set of islet-expressing ECM proteins.

Response: We thank the reviewer for this interesting suggestion. The main concern we have though is that our proteomic data is from whole pancreas tissue and our goal is to characterize the broad changes of the tissue as a whole, and by broad compartments of islet vs. acinar, while focusing on how the environment of the tissue changes as it develops. Therefore, it would not be fair or accurate to compare our results with the many RNAseq or scRNAseq studies that have been published recently, which usually focus on specific types of cells since the same protein may be expressed in multiple cell types. In addition, most of these studies used isolated islets or *in vitro* cultured islets which are not always pure and therefore often contain a substantial fraction of acinar tissue as well. Cross et al. demonstrate that at the protein level, islet ECM is also severely damaged during islet isolation and importantly, several ECM proteins are not regenerated during islet culture (PMID: 27456745).

Despite all of these concerns, we do agree that valuable information about islet ECM could be extracted from such a comparison. We compared the matrisome data from the adult donors (average of all younger (Y) and older (O) adults, N=12) to a set of unpublished RNA-seq data from cultured adult islets (N=3) generated in our own lab. Of interest, all 117 ECM-associated proteins we identified in the adult donors exhibited detectable gene expression in isolated islets. To compare relative gene-specific expression among the two studies, we ranked the 61 identified Core Matrisome genes in the two lists, using the iBAQ intensities for the proteome data (shown below). Comparison of the ranked lists enabled a general assessment of which ECM genes are more highly expressed by isolated islets, and may help to identify genes that are enriched in islets compared to acinar regions. For example, isoforms of collagen 1 and collagen 6 rank highly in both lists, which is unsurprising considering that these are known to be abundant proteins in both the islets and the whole pancreas. A few proteoglycans (DCN, LUM, OGN) rank very highly in abundance in the pancreas total protein, but rank very low in isolated islet RNA expression. This could mean that these genes are not expressed in islets, but it could also mean that regulation of these genes is disrupted during the process of islet isolation and culture. For these reasons, we determined that adding details of this comparison into the manuscript does not meaningfully contribute to our analysis. But we have added the following sentences to the manuscript to briefly mention this comparison: “In a separate unpublished study, all 117 ECM-associated genes that were identified in our proteomic study were also detected at the RNA level in isolated adult human islets, which may drive future efforts to quantify the islet-specific ECM and investigate islet-specific changes throughout development.”

Our intention to address these questions and to survey a large set of islet-specific ECM proteins is to utilize laser-capture microdissection mass spectrometry (LCM-MS) or mass spectrometry imaging (MSI) techniques to characterize the proteome of human islets *in situ* and to investigate the localization of certain proteins in the pancreas, without the need for islet isolation. This ambitious set of experiments will be conducted soon following our current study.

Ranked by expression within each study, sorted by iBAQ Rank.								
Gene Symbol	iBAQ Rank	RNASeq Rank		iBAQ Rank	RNASeq Rank		iBAQ Rank	RNASeq Rank
COL1A1	1	1	TINAGL1	22	28	COL5A1	43	16
COL6A1	2	18	FGA	23	44	CRELD2	44	38
COL1A2	3	5	TGFBI	24	26	LAMC3	45	29
COL6A2	4	11	MFAP2	25	50	PCOLCE	46	46
COL6A3	5	2	COL12A1	26	32	COL4A1	47	6
DCN	6	45	COL18A1	27	9	FBLN2	48	54
LUM	7	36	AGRN	28	22	MFGE8	49	37
OGN	8	60	LAMB2	29	24	COL2A1	50	61
FBN1	9	30	PRELP	30	49	IGFBP2	51	21
BGN	10	13	VTN	31	53	FGL1	52	51
LAMC1	11	12	VWA1	32	17	TNXB	53	57
HSPG2	12	10	LAMA2	33	39	COL4A3	54	56
COL14A1	13	43	LAMA5	34	15	COL5A3	55	35
NID1	14	23	FBN2	35	58	COL16A1	56	42
EMILIN1	15	40	COL15A1	36	20	POSTN	57	33
NID2	16	41	IGFBP7	37	19	ASPN	58	55
LAMB1	17	14	LAMA4	38	27	FRAS1	59	34
COL3A1	18	4	COL4A2	39	7	THBS1	60	8
FN1	19	3	VWF	40	31	FBLN1	61	52
FGB	20	48	COL5A2	41	25			
FGG	21	47	COL11A1	42	59			
								highest expression
								lowest expression

II. Other points:

1. In Fig 2b, no protocol seems to be provided for how to select the enriched biological processes and pathways. FDR and ranking need to be indicated for each process.

Response: We thank the reviewer for this critical comment and we have added the following sentences to the “Methods” section to describe how to enrich biological processes and pathways and select the terms that are displayed in **Fig. 2b**. “For protein intensity profiling, biological processes and pathways were enriched using DAVID bioinformatics resources (PMID: 19131956) with a FDR cutoff of 0.05. Four selected terms with lower *p*-values were shown for each cluster and the same term was only shown once where it appeared the most significant.” Only three terms were therefore shown for the 3rd cluster because the remaining terms enriched in this cluster were either above the cutoff threshold or more significant in other clusters. We also provided the detailed protein identities for each cluster, *p*-values and ranking for each term in the “Source Data” under sheet “Fig. 2”.

2. The identities of the significantly changed proteins, log₂(fold change), and p value need to be shown as a table (or within Table 3 and 5) for in Fig 2c and Fig 3b, respectively, so that readers can sort the data.

Response: We appreciate this suggestion from the reviewer, and we have added the identities, log₂(fold change), and *p*-values for all quantified proteins while highlighting the significantly changed ones. These are shown in **Supplemental Table 3** for all proteins corresponding to **Fig. 2c**, and in **Supplemental Table 5** for ECM proteins corresponding to **Fig. 3b**. We hope that this information can help the reviewer and our readers easily sort the data and extract useful information at their discretion.

3. In Table 3: Is there information about total precursor ion intensity for each protein, which can be split into intensities for that protein in each sample by their $\log_2(\text{intensity})$? The abundance information is missing from the entire analysis. It is important to know the ECM compositions and the more abundant ECM proteins and whether there are over-abundant peptides/proteins that take up most of the MS spectra.

Response: We thank the reviewer for the critical comment. In this study we used an isobaric labeling strategy to achieve a multiplexing analysis so the precursor ion intensity for each protein was a combination of multiple samples from different age groups. Therefore, this method provides accurate comparative results of the same protein across different groups but does not lend itself to provide accurate age-dependent abundance information of various proteins. Yet, we agree it would be interesting to know the protein/ECM composition in the samples. In the original manuscript, we did provide some abundance information such as in **Supplementary Fig. 1** where we showed the iBAQ (Intensity Based Absolute Quantification) intensities for proteins identified in young and older adult groups. We have slightly revised this figure by highlighting all quantified ECM proteins as shown below (red circles). All protein intensities were also provided in “**Source Data**” under sheet “SI Fig. 1”. Notably, we were able to observe some high-abundance ECM proteins as expected, such as COL1A1 and FBN1. Though there were some over-abundant non-ECM proteins, the intensities of the ECM proteins still spanned near five orders of magnitude, further highlighting the effectiveness and superiority of our methods in detecting the ECM-inclusive proteome.

Updated Supplementary Fig. 1:

Supplementary Fig. 1 Dynamic intensity range of identified proteins in adult groups. Proteins identified in young and older adult groups are ranked and plotted from high to low based on iBAQ (Intensity Based Absolute Quantification) intensities. All quantified ECM proteins are highlighted in closed red circles. Detailed information is provided in “**Source Data**”.

4. In Table 3: Additional information is needed: percent sequence coverage; total number of peptides (i.e., peptide sequence matches) assigned; number of unique peptide sequences; number of spectra.

Response: We thank the reviewer for this great suggestion, and we have added all of this additional information in **Supplementary Table 3**.

5. In Fig S9a, some of the artery signal (the yellowish elastic fiber signal) seems to be autofluorescence. A no primary antibody control will help. Scale bars are missing.

Response: We thank the reviewer for these comments and suggestions. The IgG isotype controls clearly indicate that there is a very low level of autofluorescence in the arteries and vessels compared to the positive staining we see with some of the protein-specific antibodies. We have added an IgG isotype control stain

to the panel in **Supplementary Fig. 11a**, as well as a CD31 (red) and insulin (green) co-stain, to show the lack of green autofluorescence in the vessels and ducts, and to also more clearly identify the vessels with CD31 as a marker. We think these control images would help to clarify the validity of the positive staining in the vessels and ducts with COL14A1, FN1 and POSTN in the adult tissues. We have also added a scale bar to the panel.

Updated Supplementary Fig. 11:

Supplementary Fig. 11 Localization of ECM proteins in specific regions of the pancreas. **a)** Immunofluorescent staining for ECM proteins (FN1, COL14A1, POSTN = green) which had very low levels of signal in islets (Insulin = red) and acinar, but relatively high total protein content. These proteins were mainly found to be expressed in vessels and ducts. Control images indicate low levels of autofluorescence in the ducts (white arrows) and vessels (identified in red with positive CD31 staining; yellow arrows) when stained with an insulin antibody in green, or when stained with an IgG isotype control. **b)** Immunofluorescent staining of endocrine markers (GCG, alpha cells) (SST, delta cells) indicating the at FBN2 co-localizes with delta cells, in both fetal and adult islets. Scale bars = 100 microns. Localization of ECM proteins in specific regions of the pancreas.

6. Zoomed-in images on the ECM IF staining will help readers to decide compartmental (intracellular vs. extracellular) signal.

Response: We thank the reviewer for this suggestion, as the higher magnification images do make it easier to see these trends in protein localization. We have added higher magnification images for each ECM staining in **Supplementary Fig. 9**.

Updated Supplementary Fig. 9:

Supplementary Fig. 9 Cellular localization of ECM proteins in pancreatic islets. Enlarged images of immunofluorescent staining for ECM proteins within human pancreatic islets. Images represent either fetal (F) or young adult (Y) donors as indicated, selected based on which age had higher intensity staining for each protein. Images clearly show differences in subcellular localization; most ECM proteins are expressed extracellularly while some proteins (such as EMILIN1 and FBN2) appear to be expressed intracellularly. Scale bar = 50 microns.

7. In Fig 3b, what is the criteria for showing the protein labels? Some of the most changed dots are not labeled, such as in JvsF and YvsF panels.

Response: We thank the reviewer for this critical comment. To ensure the figures are not too crowded and every label can be seen clearly, we only annotated core matrisome in **Fig. 3b** which was also mentioned in the caption of **Fig. 3**. We used the way that Naba et al. proposed to define “matrisome” (PMID: 22159717) which includes all components constituting the extracellular matrix (the “core matrisome”) and those components associated with it (“matrisome-associated” proteins). Most of our analyses and discussions focused on core matrisome including ECM glycoproteins, collagens, and proteoglycans. Matrisome-associated proteins include ECM-affiliated proteins, ECM regulators and secreted factors which could also be interesting to look at but are not the main focus of this study. For instance, among the most significantly changed proteins in juvenile vs. fetal (J vs F) and young adult vs. fetal (Y vs F) is SERPINH1 (collagen binding protein), REG1A (associated with pancreas regeneration), and CELA2A (elastase that enhances insulin signaling). We provided detailed information in the supplementary tables as a resource for readers who are interested in these matrisome-associated proteins.

Reviewer #2 (Remarks to the Author):

The manuscript by Li et al. describes the proteomic composition of 23 human pancreas samples across 4 age groups to reveal insight into development and aging associated changes. They employ their diLeu 12plex quantification and SCAD sample preparation methods to obtain very good proteome coverage with a focus on the matric component. Imaging is used to provide context for many of the ECM proteins which provides an excellent resource for regenerative medicine efforts aimed at in vitro and in vivo islet/beta cell therapies. The manuscript is well written, the figures and data also to be of high quality further supporting the importance of this work. There are some limitations to the study that need to be addressed but overall this is a valuable contribution to the field.

Response: We truly appreciate the positive evaluation from the reviewer and the praise on our work. We agree that the results presented in this study can provide guidance for future studies in the field and serve as a good resource for readers with different backgrounds. And we thank the reviewer for the careful examination of our manuscript to help us enhance some details. We have corrected or clarified several points as requested below.

Introduction

Line 40 - “but until recently, few studies have been able to extensively define the matrisome of human tissues⁸.”

This is not the case there have been over 40 examples to date. Please tone down this claim.

Response: We thank the reviewer for this comment. We changed the wording of this line to emphasize that recent studies have indeed begun this line of research: “but only recently have studies been able to extensively define the matrisome of human tissues, and studies of ECM composition throughout human development are even more limited”.

Results

Line 125 – it is mentioned that data spans six orders of magnitude. The figure referred to indicates the iBAQ method was used to estimate protein abundance. This is not described in the methods, please add.

Response: We thank the reviewer for this helpful suggestion and we have added the following sentences in the Methods to describe how to use the iBAQ method: “To compare abundance of different proteins, iBAQ method was used embedded in MaxQuant (version 1.5.2.8). Raw files were searched against the Uniprot *H. sapiens* reviewed database (September 2018) with trypsin/P selected as the enzyme and three missed cleavages allowed. DiLeu labeling on peptide N-termini and lysine residue (+145.12801), and carbamidomethylation of cysteine residues (+57.02146 Da) were chosen as fixed modifications. Variable modifications included oxidation of methionine residues (+15.99492 Da), deamidation of asparagine and

glutamine residues (+0.98402 Da) and hydroxylation on proline residues (+15.99492 Da). The iBAQ feature was enabled and all other parameters were set as default.”

Line 195 – please perform quantitative analysis on the images and report. It is not clear if image analysis was only performed on a representative sample from each age group. Please indicate in the methods at a minimum.

Response: We thank the reviewer for sharing this concern. We clarified this sentence: “Qualitatively, the immunofluorescence staining correlates well with and validates the matrisome data.” The subsequent section of the manuscript details the quantitative image analysis we performed. Our methods state that 3 donors were quantified for each age for each ECM stain (“images were taken for N=3 donors in each developmental group”). We also added to the figure legends: “Representative images are shown, images were taken for N=3 donors per developmental group.”

Discussion

Line 356 – “comprehensive”. It is not clear that this is a comprehensive analysis (this is very challenging to determine!). consider revising this term. In addition, the sample preparation included a water wash that was discarded. It is almost certain that this fraction contained a significant protein fraction.

Response: We thank the reviewer for this comment. We have removed the word ‘comprehensive’ from this sentence. The water wash is needed to remove some floating lipids and we agree a small proportion of the sample may be lost. However, the solubility of most proteins is relatively low in pure water and ECM proteins which are of special interest are even less soluble. We performed this water wash quickly (3 seconds) to minimize any possible sample loss.

Methods

Line 380 – It is an important point that these are not necessarily “normal” tissues collected under ideal conditions.

Response: We thank the reviewer for sharing these concerns and we hope to provide clarification. From the perspective of the pancreas, we used tissue from healthy pancreata with no signs of endocrine or exocrine dysfunction (i.e. no history of diabetes, normal HbA1c, normal pancreatic enzymes including amylase and lipase). The organ harvest was performed in accordance with standard protocols for organ transplantation, and organ processing was completed rapidly following organ harvest. This ensures that the tissue was collected and processed as quickly as possible and was consistent with a timeline that would be used if the organs were transplanted. Therefore, the pancreata are considered viable and healthy.

Line 396 – It is not clear when SDS solution was added. Please add enough detail necessary for others to follow the methods without referring to your previous work.

Response: We thank the reviewer for this comment. In the original manuscript we mentioned SDS earlier in the same paragraph as “Each sample was dissolved in 150 μ L of buffer solution (4 % SDS, 50 mM Tris buffer)”. To make this clearer, we have slightly revised these sentences as “Each sample was dissolved in 150 μ L of extraction buffer solution (4 % SDS, 50 mM Tris buffer)...Samples were centrifuged at 14,000 g for 15 min after which supernatant containing SDS (in the extraction buffer) was discarded.”

Line 418 – Earlier in the text it was stated that only 4 fractions were run. Please mention and describe here.

Response: We thank the reviewer for this comment. In the original manuscript we mentioned this in the last sentence of the same paragraph as “Fractions were collected every 4 min and concatenated into four before dried in vacuo.” To make this clearer, we have slightly revised these sentences as “Fractions were collected every 4 min and non-adjacent fractions were concatenated into four samples before being dried in vacuo for LC-MS/MS analysis.”

Line 445 – Why were peptides that did not contain all 12 report ions discarded. What is the rationale and describe here if appropriate.

Response: We thank the reviewer for the careful examination and this critical comment. We are sorry to make this mistake as all peptide-spectrum matches (PSMs) are considered in this study. In some studies, especially quantitation performance evaluation using samples with known ratios, PSMs that do not contain all reporter ions are usually discarded to enable further calculation but this is not the case in our work. Most PSMs showed all reporter ions and missing intensities in some PSMs are likely due to low abundance measurements of those channels. To make sure all channels have valid values for further statistical analysis, we used the “replace missing values from normal distribution” feature in Perseus. We have deleted the sentence the reviewer indicated and added the following sentence to explain: “Missing intensities were replaced using the “replace missing values from normal distribution” feature in Perseus (version 1.6.0.7) prior to further processing.”

Tables and figures – list the values used for quantification. TABLES 3, 5, 6 All have values that are some log transformed value.

Response: We thank the reviewer for this comment, and we would like to provide some explanations. These values are log₂ transformed DiLeu reporter ion intensities and we intended to show them in the tables. For most of our figures, we used log₂(intensity) or log₂(fold change) so we think it would be easier for our readers to repeat the analysis or do other tests at their discretion. Transformation is an accepted necessary step prior to many analyses. In comparative experiments, non-transformed data is usually substantially right skewed: half of the data-point are between 0 and 1 (with 1 meaning no change), and the other half between 1 and positive infinity. However, a normal or Gaussian distribution is often regarded as ideal as it is assumed by many statistical methods. The ratios become symmetrical around 0 when log transformed and therefore, a parametric statistical test provides a more accurate and relevant approach. Furthermore, a log transformed data display allows a broader set of values to be displayed on the same graphic.

Table 6 - COL1A1 value is 14.57 and COL1A2 is 14.28. these two proteins are usually very close to a 2:1 ratio. Please explain if this is iBAQ log₁₀ and what the corresponding measured ratio is.

Response: We thank the reviewer for this comment. These are DiLeu reporter ion intensities (log₂) for relative abundance comparison of the same protein from different samples. So, they are probably not very accurate to compare different protein abundances. The iBAQ intensities were provided in the “**Source Data**” under sheet “SI Fig. 1”. The intensity of COL1A1 is 7.72E+08 and COL1A2 is 4.30E+08 so indeed they show a ratio of 1.8 which is close to the theoretical value of 2. The slight deviation may result from the sample loss during sample preparation or different ionization efficiency in the mass spectrometer.

Reviewer #3 (Remarks to the Author):

Overall, this is an interesting study conducted by an expert team of investigators.

The authors present proteomic data generated using a custom-developed N,N-dimethyl leucine (DiLeu) isobaric tagging approach in samples of human pancreas of four age groups: fetal, juvenile, young adults and older adults. They identify over 3,500 proteins that change over the course of pancreas development and maturation in these different age groups, including 185 ECM proteins, of which they quantify 117. Some of the ECM proteins were also visualized by immunofluorescence in pancreatic sections.

As a whole, the study adds some new information to other publications on this topic. Data presented are interesting and are likely to inform the research community on future research efforts focusing on the function of individual ECMs in pancreas development.

Response: We are very grateful of the reviewer’s positive attitude towards our study and we agree our results can provide a good resource on future research for many readers and contribute to the community in many aspects. We also really appreciate the reviewer’s constructive comments and suggestions which

will of course help strengthen our manuscript. We have included more data, performed some additional experiments, and addressed these comments as the point-to-point responses shown below.

One of the weaknesses is that the authors make little effort to present their data in a more palatable way for non experts in the field of proteomics. For example, what is the difference between protein signals shown in the vertical columns in panels a and b of Figure 2? What is the actual identity of protein measured? What is the identity of proteins that are significantly changed between age groups shown in the Volcano plots in panel C of Figure 2? The link provided to access the raw data in the ProteomeXchange Consortium is not particularly helpful for biologists who are non experts in proteomics.

Response: We thank the reviewer for sharing this concern. We agree that the raw MS data deposited in ProteomeXchange are more useful for readers who are familiar with proteomics. Therefore, to make the dataset easier to mine for non-specialists as well, we also included all of the information in the supplementary tables. During this revision, we have included some additional lists to make them as complete as we can. We have also provided a “Source Data” file in which we provided the data used to generate many of the figures. For example, **Fig. 2a** shows the intensities of all quantified proteins and the related information (e.g. protein identity, accession, intensities) is provided in **Supplementary Table 3**. **Fig. 2b** shows the intensities of proteins that are significantly changed among different age groups (ANOVA, FDR 0.05) and these proteins are highlighted in **Supplementary Table 3**. The identity, fold change, and *p*-value for significantly changed proteins in **Fig. 2c** and **Fig. 3b** are also provided in **Supplementary Table 3** and **5**, respectively. We believe this additional information will help resolve the challenges for non-experts in MS or proteomics to view, mine or re-use our data.

We have also made an effort to make our dataset more accessible to a broad audience by creating a website which we call “Matrisome and proteome database of human pancreas” (<https://nc-webapp.herokuapp.com/>). This application contains all the key information in our manuscript and allows for custom searching of proteins or genes of interest. Below is a screenshot of the searching interface. Users will be able to view all related information (e.g. if it is an ECM protein, what ECM category it is, if it is significantly changed, fold change and *p*-value for pairwise comparisons between age groups). The full dataset is available as spreadsheets to download and we provided our contact information should people have any questions and concerns.

Accession	Gene name	Description	Is ECM protein	ECM category	Significant among age groups (ANOVA, FDR 0.05)	J vs F log2 Fold Change	J vs F pvalue	Y vs F log2 Fold Change	Y vs F pvalue	O vs F log2 Fold Change	O vs F pvalue	Y vs J log2 Fold Change	Y vs J pvalue	O vs J log2 Fold Change	O vs J pvalue
P02452	COL1A1	Collagen alpha-1(I) chain	Y	Collagens	N	0.0999	0.7004	-0.1658	0.6858	-0.1025	0.8385	-0.2657	0.5633	-0.2024	0.1

Below the table, it indicates "Showing 1 to 1 of 1 entries (filtered from 2,064 total entries)" and includes "Previous" and "Next" navigation buttons. At the bottom, there are two search input fields: "Custom accession search" and "Custom gene name search", each with a "Search" button. A "Reset" button is also present.

Another limit of the study is that there is no effort to sort out ECM (or other proteins) that are enriched in pancreatic islets. While it is understood that this was done using whole pancreata, the information presented may prove of difficult application to the study of some of these ECMs in islet biology. Some notions about ECMs possibly implicated in islet development and function can only be inferred by a derivative comparison with other published studies (see Naba et al., Scientific reports 7, 40495, 2017).

Response: We thank the reviewer for sharing this concern, and we recognized this could be a barrier. We agree that the whole-tissue mass spectrometry methodology we employed cannot be used exclusively to make conclusions about ECM and proteins relevant specifically to islet development. While our main goal is to characterize the broad changes of the tissue as a whole while focusing on how the environment of the tissue changes as it develops, we also think it is important to sort out ECM proteins enriched in islets. For this reason, we used IF staining to visualize the localization of some ECM proteins in islets compared to acinar regions throughout development (**Fig. 5**).

Obviously, to survey a larger set of ECM proteins in the islets, alternative strategies are needed and we have thought of investigating isolated islets or comparing with previous publications. However, isolated islets are not always pure and therefore often contain a substantial fraction of acinar tissue as well. Effective efforts to isolate islets from the human pancreas inherently destroy the ECM and several ECM proteins are not regenerated during islet culture (Refs: PMID: 27456745, PMID: 32852857). The 2017 Naba report we cited and compared with our data utilized an islet isolation technique that was not translatable to the much larger and denser human pancreas. It is also difficult to compare our proteomic data from whole pancreas tissue with many previous studies (often RNAseq) which usually focus on specific types of cells since the same protein may be expressed in multiple cell types.

Despite these concerns, we do agree that valuable information about islet ECM could be extracted using data from isolated islets. We therefore compared the matrisome data from the adult donors (average of all younger (Y) and older (O) adults, N=12) to a set of unpublished RNA-seq data from cultured adult islets (N=3) generated in our own lab. Of interest, all 117 ECM-associated proteins we identified in the adult donors had detectable gene expression in the isolated islets. To compare relative gene-specific expression among the two studies, we ranked the 61 identified Core Matrisome genes in the two lists, using the iBAQ intensities for the proteome data (shown below). Comparison of the ranked lists enabled a general assessment of which ECM genes are more highly expressed by isolated islets, and may help to identify genes that are enriched in islets compared to acinar tissue. For example, isoforms of collagen 1 and collagen 6 rank highly in both lists, which is unsurprising considering that these are known to be abundant proteins in both the islets and the whole pancreas. A few proteoglycans (DCN, LUM, OGN) rank very highly in abundance in the pancreas total protein, but rank very low in isolated islet RNA expression. This could mean that these genes are not expressed in islets, but it could also mean that regulation of these genes is disrupted during the process of islet isolation and culture. For these reasons, we believe that adding this comparison into the manuscript does not meaningfully contribute to our analysis.

To address these questions about islet-specific ECM content, we intend to conduct our own future study using laser-capture microdissection mass spectrometry (LCM-MS) or mass spectrometry imaging (MSI) techniques to characterize the proteome of human islets *in situ* and to investigate the localization of certain proteins in the pancreas, without the need for islet isolation. This will avoid the inherent damage islets undergo during the process of isolation and culture, and give much more meaningful data regarding islet development in healthy human tissues. However, this ambitious project is beyond the scope of our current study.

Ranked by expression within each study, sorted by iBAQ Rank.								
Gene Symbol	iBAQ Rank	RNASeq Rank		iBAQ Rank	RNASeq Rank		iBAQ Rank	RNASeq Rank
COL1A1	1	1	TINAGL1	22	28	COL5A1	43	16
COL6A1	2	18	FGA	23	44	CRELD2	44	38
COL1A2	3	5	TGFBI	24	26	LAMC3	45	29
COL6A2	4	11	MFAP2	25	50	PCOLCE	46	46
COL6A3	5	2	COL12A1	26	32	COL4A1	47	6
DCN	6	45	COL18A1	27	9	FBLN2	48	54
LUM	7	36	AGRN	28	22	MFGE8	49	37
OGN	8	60	LAMB2	29	24	COL2A1	50	61
FBN1	9	30	PRELP	30	49	IGFBP2	51	21
BGN	10	13	VTN	31	53	FGL1	52	51
LAMC1	11	12	VWA1	32	17	TNXB	53	57
HSPG2	12	10	LAMA2	33	39	COL4A3	54	56
COL14A1	13	43	LAMA5	34	15	COL5A3	55	35
NID1	14	23	FBN2	35	58	COL16A1	56	42
EMILIN1	15	40	COL15A1	36	20	POSTN	57	33
NID2	16	41	IGFBP7	37	19	ASPN	58	55
LAMB1	17	14	LAMA4	38	27	FRAS1	59	34
COL3A1	18	4	COL4A2	39	7	THBS1	60	8
FN1	19	3	VWF	40	31	FBLN1	61	52
FGB	20	48	COL5A2	41	25			
FGG	21	47	COL11A1	42	59			
								highest expression
								lowest expression

The expression of some of the ECM identified in the study was validated in situ for only a handful of them, mainly collagens and laminins (which incidentally were already known to be expressed at different levels in different compartments of the pancreas). Osteoglycin (OGN) and periostin (POSTN) are perhaps the only two examples of proteins that have been studied less in the pancreas.

Response: We thank the reviewer for this feedback, but we do not completely agree with these statements. The existing information about ECM in the pancreas and islets has been collected using tissues from a variety of different species, and the results are not always consistent with one another. Ours is the first study to use human pancreas at pre-natal and multiple post-natal developmental stages, and use mass spectrometry to comparatively quantify ECM levels. Previous literature has studied individual proteins in a small number of human donors at relatively few distinct age ranges. The most extensive is the Otonkoski et al. 2008 study (and Vertanen et al. 2008 cited within) which we cite and discuss. They were among the first to analyze a variety of laminins in human fetal and adult tissues, but only used IF staining, and did not perform quantitative proteomic studies. For this reason, we did not focus on doing as much laminin IF staining in our study, but instead compared our proteomic results to their findings, which were consistent with one another, as reported in the second paragraph of our Discussion section. Other existing literature often implies the existence or observation of a specific ECM protein in human pancreas or islets, but references studies that include only data from non-human tissues, or that have not actually measured the specific ECM protein of interest at all. For these reasons, in our study we identified at the proteomic level, all ECM proteins in the pancreas and focused our immunofluorescent staining on specific ECM proteins of interest as a validation, and uniquely studied the expression by proteomics and IF staining of pre-natal and multiple post-natal developmental stages. We believe these data substantially add to the current knowledge about human pancreas ECM, especially at the younger (fetal, juvenile) developmental stages.

All of the immunolocalization experiments shown, both in Figure 4 and in the Supplemental Figures 7, 8 and 9, lack important negative isotype IgG controls. Without these controls these data have little or no value, especially considering that nowadays companies selling these antibodies provide terrible quality control assurance. Although the authors provide an Excel document where they identify all antibodies used in these immunolocalization experiments, they seem to think that the so-called “Antibody validation information” provided by vendors for each antibody is enough to guarantee the specificity of the antibodies. This is categorically not true, each immunostaining performed needs its own intra-assay IgG control. Historical control experiments conducted on a few tissue sections are not acceptable, as the condition of the tissue sections, the donors, and other variables (e.g. pH of the phosphate buffer used in a given day) may dramatically affect the outcome of the staining.

Response: We thank the reviewer for sharing these comments and agree that inclusion of our negative controls is important for interpreting the immunostaining data presented in this report. We have included representative images of staining using rabbit and mouse IgG isotype control antibodies in **Supplementary Figure 8b**. These images clearly show a lack of significant autofluorescence or non-specific binding in the pancreas parenchyma. Furthermore, the changes in protein expression and localization throughout the developmental age groups we studied also provide intrinsic controls for the antibodies. For example, COL14A1 has positive signal in the fetal tissue, and reduced signal in the postnatal tissues, while COL16A1 has strong signal in the postnatal tissues and little positive signal in the fetal tissues. These internal controls not only show that there is unstained negative space in the samples with less positive signal, but also internally validate the antibodies because these trends strongly correlate with our mass spec data for each protein.

Updated Supplementary Fig. 8:

Supplementary Fig. 8 Visualizing ECM proteins in human pancreata across developmental stages.

a) Immunofluorescent images of ECM proteins (green) co-stained with insulin (red) in fetal (F), juvenile (J), young adult (Y) and older adult (O) pancreata. Qualitative trends in protein levels corroborate mass spec data. Representative images are shown, images were taken for N=3 donors per developmental group.
b) IgG isotype control images with rabbit or mouse IgG (green) indicate low levels of non-specific signal. Scale bar = 100 microns.

Most of the Figure legends are extremely succinct, and lack important details for the reader to understand what the data show.

Response: We appreciate this feedback from the reviewer and agree that it is important to strive for clear and easy-to-understand figures. For this reason, we have reassessed all figure legends and added more information and clarification as necessary to ensure that our data are easy to understand for all readers.

REVIEWERS' COMMENTS

Reviewer #1 (Remarks to the Author):

The authors addressed most of my comments to various levels. The remaining comments are:

In response to I-2, fibronectin is not a basement membrane component.

In response to II-5, IgG control need to be shown for the same type of tissues. Please show IgG control images with artery and the major interlobular ducts.

Reviewer #3 (Remarks to the Author):

The authors have made reasonable revisions to the manuscript.

REVIEWERS' COMMENTS

Reviewer #1 (Remarks to the Author):

The authors addressed most of my comments to various levels. The remaining comments are:
In response to I-2, fibronectin is not a basement membrane component.

In response to II-5, IgG control need to be shown for the same type of tissues. Please show IgG control images with artery and the major interlobular ducts.

Response: We really appreciate these additional comments from the reviewer and have edited our manuscript accordingly. In line 321, we have removed “fibronectin” when mentioning examples for “basement membrane”. In **Supplementary Fig. 11**, we have shown IgG control images with vessels and ducts.

Updated **Supplementary Fig. 11a**:

Reviewer #3 (Remarks to the Author):

The authors have made reasonable revisions to the manuscript.

Response: We thank the reviewer again for all the helpful suggestions to improve our manuscript.